

**The haze pollution under strong atmospheric oxidization capacity in summer in**
**Beijing: Insights into the formation mechanism of atmospheric physicochemical**
**process**
Dandan Zhao[†1,2]; Guangjing Liu[†3,1]; Jinyuan Xin[*1,2,4]; Jiannong Quan[5]; Yuesi Wang[1]; Xin Wang[3];
Lindong Dai[1]; Wenkang Gao[1]; Guiqian Tang[1]; Bo Hu[1]; Yongxiang Ma[1]; Xiaoyan Wu[1]; Lili
Wang[1]; Zirui Liu[1]; Fangkun Wu[1]
1 State Key Laboratory of Atmospheric Boundary Layer Physics and Atmospheric Chemistry (LAPC), Institute of
Atmospheric Physics, Chinese Academy of Sciences, Beijing 100029, China
2 University of Chinese Academy of Sciences, Beijing 100049, China
3 College of Atmospheric Sciences, Lanzhou University, Lanzhou 730000, China.
4 Collaborative Innovation Center on Forecast and Evaluation of Meteorological Disasters, Nanjing University of
Information Science and Technology, Nanjing 210044
5 Institute of Urban Meteorology, Chinese Meteorological Administration, Beijing, China
(†) These authors contributed equally to this study.
(*) Correspondence: Jinyuan Xin (xjy@mail.iap.ac.cn)
**Abstract:** Under strong atmospheric oxidization capacity, haze pollution in the summer
of Beijing was the result of the synergistic effect of physicochemical process in the
atmospheric boundary layer (ABL). The south/southwest areas generally ~60-300 km
far away from Beijing were seriously polluted, in contrast to a clean situation in Beijing.
The southerly winds moving more than ~20-30 km h[-1] since early morning primarily
caused the initiation of haze pollution. The $PM_{2.5}$ level increased to 75 μg m[-3] in several
hours at daytime, which was simultaneously affected by the ABL structure. Additionally,
the $O_3$ concentration was quite high at daytime (250 μg m[-3]), corresponding to a strong
atmospheric oxidation capacity. Numerous sulfate and nitrate were formed through
active atmospheric chemical processes, with sulfur oxidation ratio (SOR) up to ~0.76
and nitrogen oxidation ratio (NOR) increasing from 0.09 to 0.26, which further
facilitated the particulate matter (PM) level rising. Even so, the increase in sulfate was
mainly linked by southerly transport. At midnight, the $PM_{2.5}$ concentration sharply
increased from 75 μg m[-3] to 150 μg m[-3] in 4 hours and stayed the highest level till the



next morning. With the premise of an extremely stable ABL structure, the formation of
secondary aerosols dominated by nitrate was quite intense, driving the outbreak of haze
pollution. PM levels in the south/southeast of Beijing were significantly lower than that
in Beijing over this time, even below air quality standards, thus, the contribution of
pollution transport was almost gone. With the formation of nocturnal stable boundary
layer of 0-0.3 km altitude, the extremely low turbulence kinetic energy (TKE) of 0-0.05
$m^2\ s^{-2}$ inhibited the spread of particles and moisture, ending up with elevated levels of
$PM_{2.5}$ and relative humidity (~90 %) near the surface. Under quite high humidity and
strong ambient oxidization capacity, the NOR rapidly increased from 0.26 to 0.60 and
heterogeneous hydrolysis reactions at the moist particle surface were very significant.
The nitrate concentration explosively increased from 11.6 $\mu g\ m^{-3}$ to 57.8 $\mu g\ m^{-3}$, while
the concentrations of sulfate and organics slightly increased by 6.1 $\mu g\ m^{-3}$ and 3.1 $\mu g$
$m^{-3}$, respectively. With clean & strong winds passing through Beijing, the stable ABL
was broken with potential temperature gradient turning to negative and ABL heights
increasing to ~2.5 km. The strong turbulence activity with TKE of ~3-5 $m^2\ s^{-2}$ notably
promoted the pollution diffusion. The self-cleaning capacity of the atmosphere is
always responsible for the dispersion of air pollution. Even so, reducing atmospheric
oxidization capacity such as strengthening the collaborative control of nitrogen oxide
(NOx) and volatile organic compounds (VOCs) was urgent, as well as continuously
deepening regional joint control of air pollution.
**1  Introduction**
Due to a series of stringent emission control measures (China's State Council 2013
Action Plan for Air Pollution Prevention and Control available at
http://gov.cn/zwgk/2013-09/12/), including shutting down heavily polluting factories and
replacing coal fuels with clean energies, the annual mean $PM_{2.5}$ (particulate matter with
dynamic equivalent diameter less than 2.5 $\mu m$) concentration in major regions,
especially in Beijing, has dropped continuously in recent years (Chen et al., 2019; Liu
et al., 2019a; Cheng et al., 2019a; Ding et al., 2019). However, the ground-level $O_3$
concentration across China increased rapidly in recent years, especially in summer,



despite recent reductions in the emissions of $SO_2$ and nitrogen oxide (NOx) (Chen et
al., 2018; Anger et al., 2016; Wang et al., 2018; Wang et al., 2017b). This discrepancy
of variation trend between $O_3$ and $PM_{2.5}$ may be attributed to the inappropriate reduction
ratio of NOx and volatile organic compounds (VOCs) in $PM_{2.5}$-control oriented
emission reduction measures which mainly focus on NOx reduction (Liu et al., 2013a;
Cheng et al., 2019b). Besides, a number of studies have shown that the reduce in
ambient particles can influence the surface ozone generation via changing the
heterogeneous reaction and decreasing the photodecomposition rate ($O_3$ and its
precursors) through the aerosol-radiation interaction (Liu et al., 2019b; Wang et al.,
2019b; He and Carmichael, 1999; Dickerson et al., 1997; Tie et al., 2001; Martin et al.,
2003; Tie et al., 2005). Recently, even though the $PM_{2.5}$ level in Beijing is generally
low due to stringent emission control measures, several haze pollution episodes with
alternate/synchronous high ozone concentration have still occurred in the summer of
2019. Regarding the causes of particulate matter (PM) pollution, numerous previous
studies have reported that the stationary synoptic condition, local emissions and
regional transport, adverse atmospheric boundary layer (ABL) structure and
meteorology conditions as well as the secondary aerosol formation are major factors in
the formation of haze pollution (Li et al., 2019; Sun et al., 2012; Wang et al., 2016; Liu
et al., 2019c; Huang et al., 2017; Luan et al., 2018; Han et al., 2019). Huang et al. (2017)
demonstrated that haze pollution in the Beijing-Tianjin-Hebei usually occurred when
air masses originating from polluted industrial regions in the south prevailed and is
characterized by high $PM_{2.5}$ loadings with considerable contributions from secondary
aerosols. Bi et al. (2017) stated that strong wind and vertical mixing in daytime
scavenged the pollution, and the weak wind and stable inversion layer at night favorably
accumulated the air pollutants near the surface. Zhong et al. (2018) showed that the
positive ABL meteorological feedback on $PM_{2.5}$ mass concentration explains over 70 %
of the outbreak of pollution. Zhao et al. (2019) also pointed out that the constant
feedback effect between aerosol radiative forcing and the ABL stability continually
reduced atmospheric environmental capacity and aggravated air pollution. The



dominated components of PM, including sulfate, nitrate, ammonium, and organics, are
mostly formed via the homogeneous/heterogeneous reactions of gas phase precursors
in the atmosphere (Orrling et al., 2011; Wang et al., 2016) and account for over 50 %
of the $PM_{2.5}$ mass (Wang et al., 2013; Liu et al., 2019a; Sun et al., 2015; Yao et al.,
2002). Ming et al. (2017) have proved that the contribution of secondary aerosol
formation during haze pollution episodes was much higher than before and after the
episodes.
Although the causes of heavy $PM_{2.5}$ loading were widely examined, most of these
studies referenced to haze pollution in winter and only involved in one or several key
factors. In the summer of Beijing, with strong solar radiation, $O_3$ can be quickly
formed via photochemical reactions among precursors, including volatile organic
compounds (VOCs) and nitrogen oxides (NOx), which contributes the increase in
ambient oxidizing capacity (Wang et al., 2017c; Ainsworth et al., 2012; Hassan et al.,
2013; Trainer et al., 2000; Sillman, 1999). Meteorology conditions, including solar
radiation, temperature, relative humidity, wind speed and direction, and cloud cover,
also play an important role in short-term ozone variations, further affecting the
atmospheric oxidization capacity (Lu et al., 2019; Cheng et al., 2019b; Toh et al., 2013;
Wang et al., 2017d; Zeng et al., 2018). As ozone pollution is more and more prominent,
and the ability of atmospheric oxidation is gradually stronger, the formation
mechanism of haze pollution under strong atmospheric oxidization capacity need to
be concerned. Previous studies have demonstrated that strong atmospheric
photochemical reactions in summer enhanced secondary aerosols formation and led to
the synchronous occurrence of high concentrations of $PM_{2.5}$ and $O_3$ on a regional scale
(Pathak et al., 2009; Wang et al., 2016; Shi et al., 2015). Nevertheless, the mechanisms
of how overall regional transport, ABL structure, meteorological conditions and the
formation of secondary aerosols work together to quantitatively influence the haze
pollution under strong atmospheric oxidization capacity in summer remain unclear.
Therefore, with minutely observation of air temperature and relative & absolute
humidity profiles, vertical velocity and horizontal wind vector profiles, atmospheric
backscattering coefficient (BSC) profiles and ABL heights (ABLH), as well as mass
concentration and composition of $PM_{2.5}$, aerosol optical depth (AOD) and mass
concentrations of gas pollutants including $O_3$, $SO_2$, and $NO_2$, this paper would
comprehensively explore the formation mechanism of haze pollution under strong
ambient oxidization capacity insights into atmospheric physics and chemistry, for
proposing selected recommendations for model forecast and cause analysis of
complex air pollution in the summer of Beijing.
**2 Instruments and data**

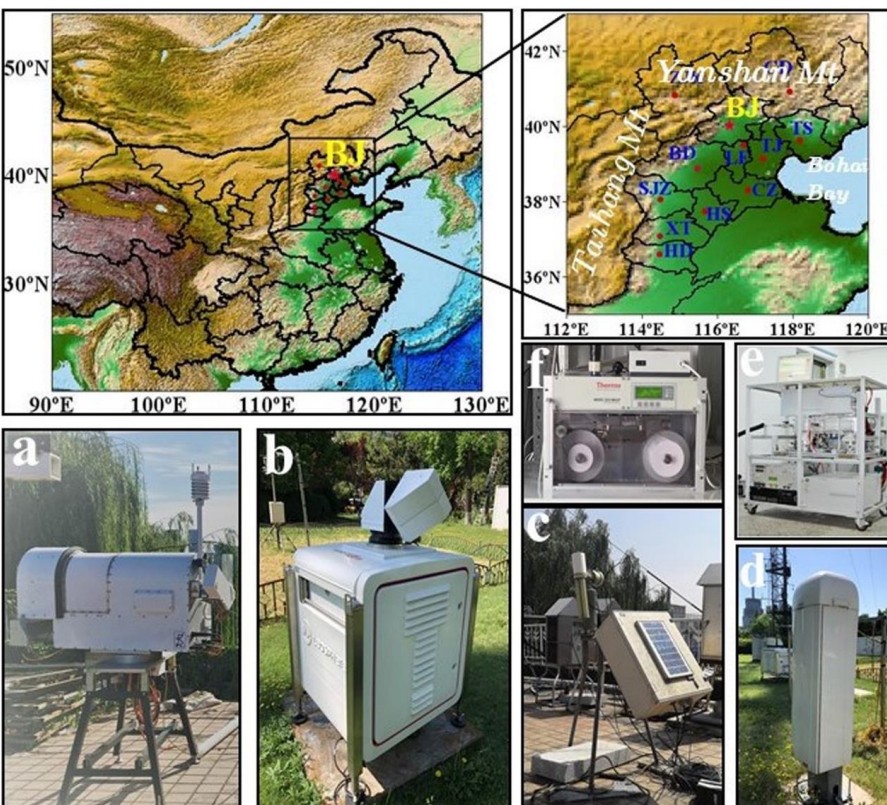

Figure 1. The geographical location of Beijing city (BJ) marked by a red star as well as
surrounding regions, and the relevant measurement instruments used in this paper. Left-top
panel is the topographic distribution of most of China with Beijing and surrounding areas
circled, and right-top panel is the topographic distribution of the Beijing-Tianjin-Hebei (BTH)
region, with Yanshan mountains to the north, Taihang Mountains to the west, and Bohai Bay to



the east. Blue words represent abbreviations for city names in the BTH. The pictures of (a)-(f)
are Microwave Radiometer, 3D Doppler Wind Lidar, CIMEL sun-photometer, Ceilometer,
Aerodyne Aerosol Chemical Speciation Monitor and Multi-angle Absorption Photometer set in
the BJ site.
**2.1 Instruments and related data**
The observation site was at the Tower Branch of the Institute of Atmospheric
Physics, Chinese Academy of Sciences (IAP, 39°58′N, 116°22′E, altitude: 58 m). And
the IAP site is located at north ring-3 and north ring-4 of Beijing, China, within
educational, commercial and residential areas, a representative for a typical urban site
in Beijing (hereinafter BJ site). All the sampling instruments are placed at the same
place and operate simultaneous monitoring. All the data used in this paper are from July
22 to 27 in 2019 and are reported in Beijing Standard Time.
Air temperature and relative & absolute humidity profiles were collected by
Microwave Radiometer (RPG-HATPRO-G5 0030109, Germany). The Microwave
Radiometer (hereinafter MWR) produces profiles at 10-30 m resolution up to 0.5 km,
40-70 m resolution from 0.5 km to 2.5 km and 100-200 m resolution from 2 km to 10
km with a temporal resolution of one second. The detailed description of instruments
of the RPG-HATPRO type can be found at the Internet site of http://www.radiometer-
physics.de.
Vertical wind speed and horizontal wind vector profiles were retrieved by 3D
scanning Doppler Wind Lidar (Windcube 100s, Leosphere, France). The wind
measurement results have a spatial resolution of 1-20 m up to 0.3 km and 25 m from
0.3 km to 3 km, with a temporal resolution of one second. More details of this
instrument can be looked up at the Internet site of www.leosphere.com.
The Ceilometer (CL51, Vaisala, Finland), is responsible for the detection of
atmospheric BSC profiles. The CL51 ceilometer digitally samples the return
backscattering signal from 0 to 100 μs and provides BSC profiles with a spatial
resolution of 10 m from the ground to a height of 15 km. As the PM is almost in the
ABL and is barely in the free atmosphere, the ABL height was determined by a sharp





change in the negative gradient in the BSC profile (Muenkel et al., 2007). More detailed
information on ABL height calculation and screening can be found in previous studies
(Tang et al., 2016; Zhu et al., 2018).
Aerosol optical depth (AOD) is observed by the CIMEL sun-photometer (CE318,
France) and AOD in 500 nm is used in this paper. CE318 is a multi-channel, automatic
sun-and-sky scanning radiometer and takes measurements only during daylight hours
(sun above the horizon). Detailed information of the AOD inversion method and the
CE318 instrument is introduced in Gregory (2011).
The real-time hourly mean ground-level of $PM_{2.5}$, $PM_{10}$, $O_3$, $NO_2$ and $SO_2$, were
downloaded from the China National Environmental Monitoring Center (CNEMC)
(available at http://106.37.208.233:20035/). All operational procedures are conducted
strictly following "The Specification of Environmental Air Quality Automatic
Monitoring      Technology"      (HJ/T193-2005,      available      at
http://kjs.mep.gov.cn/hjbhbz/bzwb/dqhjbh/jcgfffbz/200601/t20060101_71675.htm).
The chemical species of PM including organics (Org), sulfate ($SO_4^{2-}$), nitrate ($NO_3^-$),
ammonium ($NH_4^+$) and chloride ($Cl^-$) were hourly measured by Aerosol Chemical
Speciation Monitor (ACSM). More detailed descriptions for ACSM have been given in
Ng et al. (2011). The black carbon (BC) mass concentration is observed by the Multi-
angle Absorption Photometer (MAAP5012, Thermo Electron). A more detailed
description of this MAAP could be found in Petzold and Schonlinner (2004). As shown
in Fig.2, the ACSM $PM_{2.5}$ mass concentration (=organics + sulfate + nitrate +
ammonium + chloride + BC) tracked well the online $PM_{2.5}$ mass concentration, which
directly observed by the particulate matter analyzer (from the CNEMC), with the
correlation coefficient ($R^2$) of 0.82. On average, the ACSM $PM_{2.5}$ mass concentration
reports 80 % of the online $PM_{2.5}$ mass concentration. All chemical compositions
measured by the ACSM, including organics, sulfate, nitrate ammonium and chloride,
plus the BC can represent the dominant species of $PM_{2.5}$.

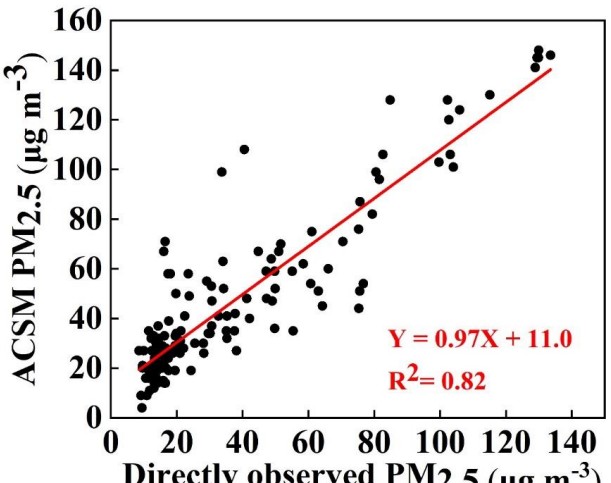


Figure 2. Scatter plot of the relationship between directly observed PM$_{2.5}$ mass concentration

by the PM analyzer from the China National Environmental Monitoring Center and calculated

PM$_{2.5}$ mass concentration from chemical constituent mass concentration measured by Aerosol

Chemical Speciation Monitor plus black carbon mass concentration measured by Multi-angle

Absorption Photometer.

**2.2 Other datasets**

Virtual potential temperature ($\theta_V$) and pseudo-equivalent potential temperature

($\theta_{se}$) are respectively calculated by Eq. (1) and Eq. (2):

$$\theta_v = T(1 + 0.608q)(\frac{1000}{P})^{0.286} \tag{1}$$

$$\theta_{se} = T(\frac{1000}{P})^{0.286} exp\left(\frac{r_s L_v}{C_{pd}T}\right) \tag{2}$$

where $T$ is air temperature, $q$ is specific humidity, $p$ is air pressure, $r_s$ is saturation

mixing ratio, $Lv$ is the latent heat of vaporization of $2.5 \times 10^6$ J kg$^{-1}$ and $C_{pd}$ is the specific

heat of air of 1005 J kg$^{-1}$ K$^{-1}$. All the relevant parameters can be calculated from

temperature and humidity profile data of MWR, then the values of $\theta_v$ and $\theta_{se}$ at

different altitudes can be obtained further. Hourly turbulence kinetic energy (TKE) is

calculated as:

$$\text{TKE} = 0.5 \times (\delta_u^2 + \delta_v^2 + \delta_w^2). \tag{3}$$

The one-hour vertical velocity standard deviation ($\delta_w^2$) and the one-hour horizontal

wind standard deviation ($\delta_u^2$; $\delta_v^2$) are respectively calculated by Eq. (4) and Eq. (5)-(6):



$\delta_w^2 = \frac{1}{N-1}\sum_{i=1}^{N}(w_i - \bar{w})^2$        (4)
$\delta_u^2 = \frac{1}{N-1}\sum_{i=1}^{N}(u_i - \bar{u})^2$        (5)
$\delta_v^2 = \frac{1}{N-1}\sum_{i=1}^{N}(v_i - \bar{v})^2$        (6)
where N is the record number every one hour, $w_i$ denotes the $i_{th}$ vertical wind velocity
(m s$^{-1}$), $u_i(v_i)$ denotes the $i_{th}$ horizontal wind speed (m s$^{-1}$), $\bar{w}$ is the mean vertical
wind speed (m s$^{-1}$), and $\bar{u}\,(\bar{v})$ is the mean horizontal wind speed (m s$^{-1}$) (Wang et al.,
2019a; Banta et al., 2006). Atmospheric reanalysis data from the National Centers for
Environmental Prediction (NCEP) were collected 4 times daily at 0200, 0800, 1400,
and 2000 (LT) with a horizontal resolution of 2.5° × 2.5°.
**3 Results and discussion**
**3.1 Typical air pollution episodes in summer in Beijing**

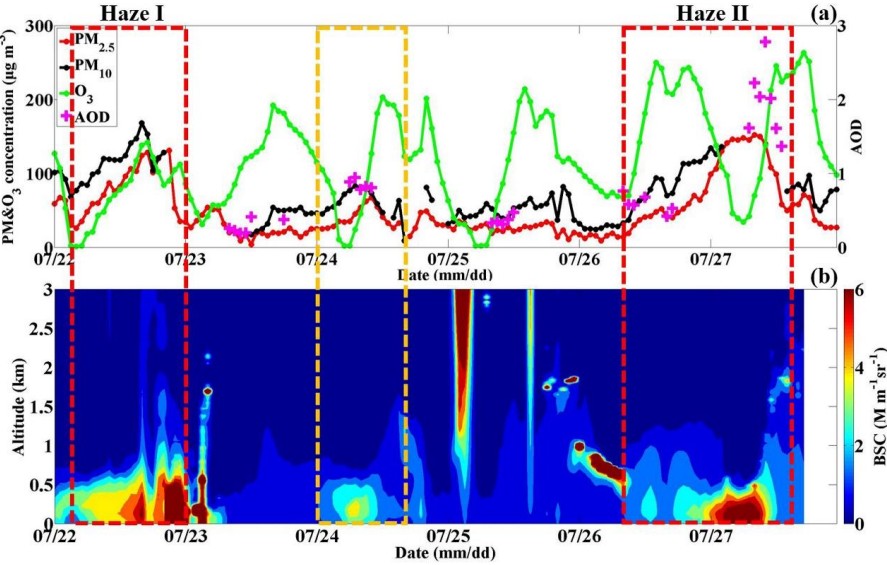


Figure 3. (a) Temporal variation on mass concentrations of PM$_{2.5}$, PM$_{10}$ and O$_3$ as well as
aerosol optical depth (AOD) in the BJ site during July 22-27, 2019; (b) Temporal variation on
vertical profiles of atmospheric backscattering coefficient (BSC) in the BJ site during July 22-
27, 2019. (The yellow-mark part represents the light haze pollution period and the red-mark
part represents heavy haze pollution episode.)

Considering both the daily-mean PM$_{2.5}$ mass concentration on the 22$^{nd}$ and 26$^{th}$-





$27^{th}$ exceeded the national secondary standard (75 μg m$^{-3}$) (GB3095-2012) with
maximum hourly average up to 131 μg m$^{-3}$ and 152 μg m$^{-3}$, respectively, two serious
PM pollution processes occurred, defined as Haze I and Haze II, respectively. During
the two haze periods, high atmospheric BSC levels mainly distributed below 0.5 km
altitude, with values of ~4-6 M m$^{-1}$ sr$^{-1}$. It reflects the vertical distribution of ambient
particles from the aspect of aerosol scattering to some degree, that is, suspended
numerous particles just concentrated in the lower atmosphere layer. According to
National Ambient Air Quality Standards (GB3095-2012), a day when the hourly-mean
$O_3$ concentration more than 160 μg m$^{-3}$ is regarded as ozone pollution day, thus, there
was serious ozone pollution every day during the observation periods. As reported by
the Ministry of Ecology and Environment, in 2018, the number of motor vehicles
reached   327   million,   up   5.5   percent   year-on-year   (available   at
http://www.mee.gov.cn/xxgk2018/xxgk/xxgk15/201909/t20190904_732374.html).
Although stringent pollution control measures on factories, the number of motor
vehicles still discharged a large number of primary pollutants into the atmosphere,
including NOx, HC, VOCs, and CO. And with strong solar radiation and high
temperature in summer, photochemical processes are prominent, contributing to the
high concentration of $O_3$ along with many highly reactive radicals, which further
enhanced the oxidizing capacity of the atmosphere (Frischer et al., 1999; Sharma et al.,
2013). Haze pollution under the condition of strong atmospheric oxidation capacity was
thought to respectively occur on the $22^{nd}$ and $26^{th}$-$27^{th}$. Generally, under stringent
pollutants emission control measures, the emission of primary aerosols is few with a
really low PM$_{2.5}$ level in summer in Beijing. The sudden elevated ambient particle
concentration (Haze I and Haze II) brought the worst PM pollution in Beijing this
summer and has been widely concerned by the public. Thus, the formation mechanism
of Haze I and Haze II in which the concentrations of PM$_{2.5}$ and $O_3$ were
simultaneously/alternately high should be discussed systematically. The key point is to
determine the oxidation capacity of regional atmosphere and to clarify the formation
mechanism of secondary aerosols. Besides, the occurrence and evolution patterns of the



two haze processes were different, which could refer to the diverse accumulation
mechanism, regional transfer contribution, ABL structure, and removal process. So, by
clarifying the various pollution processes, it is possible to conclude the leading factors
of the haze phenomenon in Beijing in summer. In short, we are going to explore the
causes of haze pollution under strong atmospheric oxidization capacity, in terms of the
physics process such as sources and sinks of pollutants and ABL structure influence,
and the chemistry process, which means aerosol transformation process.
**3.2 The formation mechanism of haze pollution in summer in Beijing**

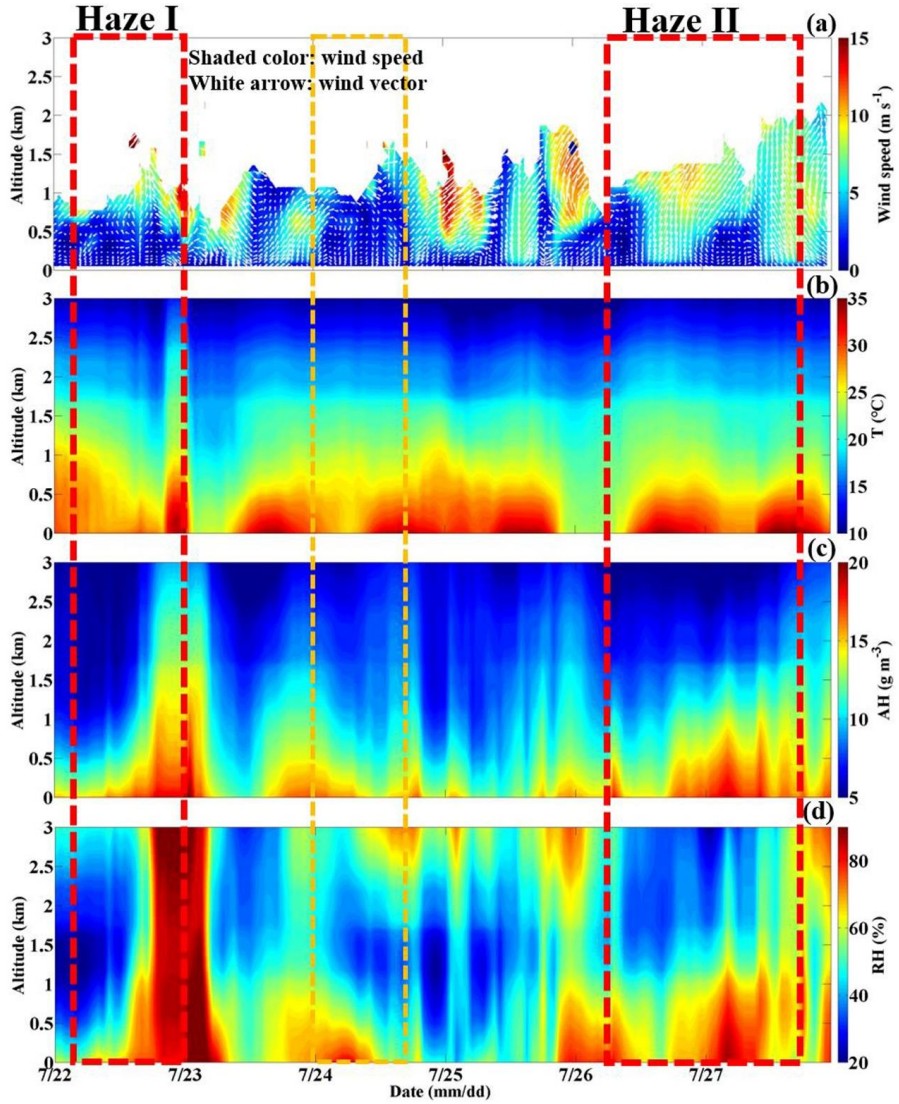

Figure 4. Temporal variation on vertical profiles of (a) horizontal wind vector (white arrows

denote wind vectors), (b) temperature (T), (c) absolute humidity (AH), and (d) relative humidity

(RH) in the BJ site during July 22-27, 2019. (The yellow-mark part represents the light haze

pollution period and the red-mark part represents heavy haze pollution episode.)

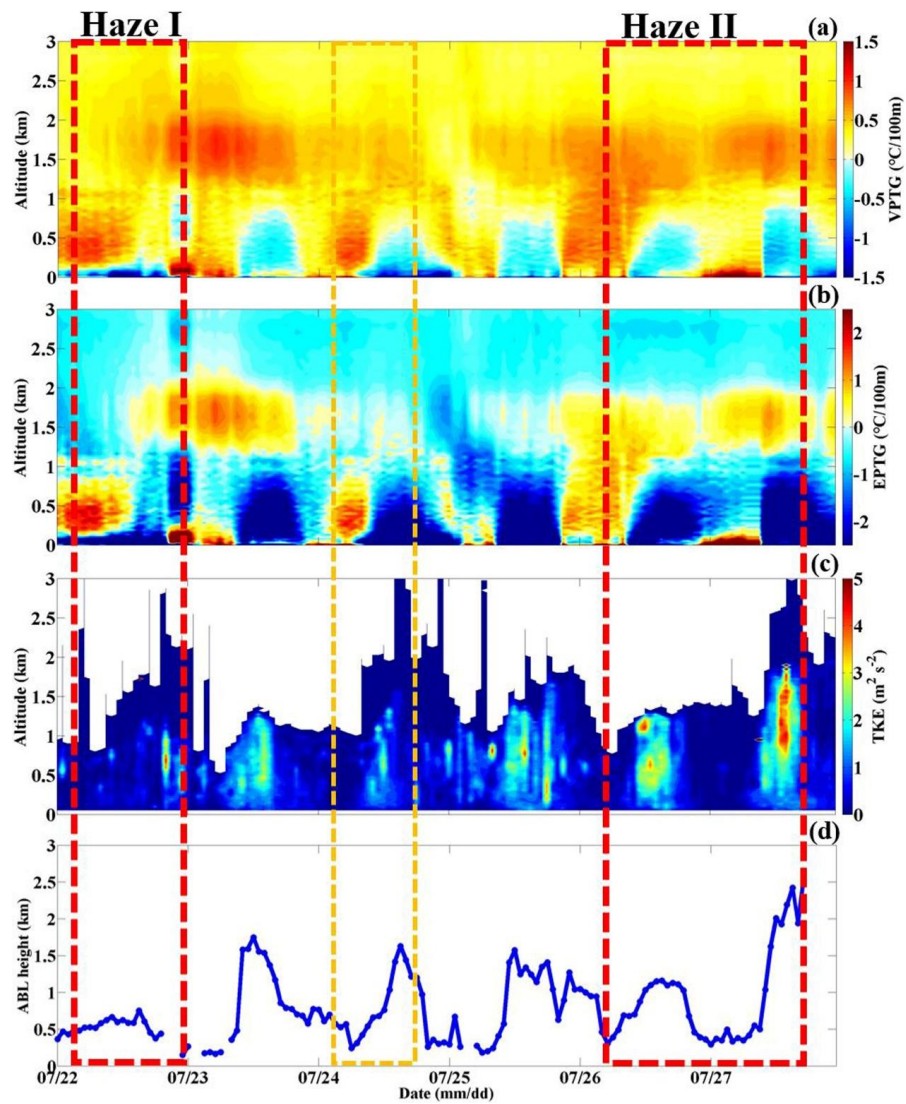

Figure 5. Temporal variation on vertical profiles of (a) virtual potential temperature gradient

(VPTG: $\partial\theta_v/\partial z$), (b) pseudo-equivalent potential temperature gradient (EPTG: $\partial\theta_{se}/\partial z$) and (c)

turbulent kinetic energy (TKE), along with corresponding (d) atmospheric boundary layer

height (ABLH) in the BJ site during July 22-27, 2019. (The yellow-mark part represents the

light haze pollution period and the red-mark part represents heavy haze pollution episode.)

Figure 6. Temporal variation on mass concentrations of (a) PM$_{2.5}$, O$_3$, NO$_2$ as well as SO$_2$, (b)



PM$_{2.5}$ chemical compositions, including Organics (Org), sulfate (SO$_4^{2-}$), nitrate (NO$_3^-$),
ammonium salt (NH$_4^+$), chlorine salt (Cl$^-$) and black carbon (BC) in the BJ site during July 22-
27, 2019. (c) Temporal variation on the relative contribution of chemical compositions to PM$_{2.5}$
mass concentration in the BJ site during July 22-27, 2019. (d) Temporal variation on mass
concentration of dominant PM$_{2.5}$ chemical compositions, sulfur oxidation ratio (SOR) and
nitrogen oxidation ratio (NOR) in the BJ site during July 22-27, 2019. (e) Temporal variation
on relative humidity (RH) and temperature (T) in the BJ site during July 22-27, 2019. (The
yellow-mark part represents the light haze pollution period and the red-mark part represents
heavy haze pollution episode.)
**3.2.1 The occurrence stage**
As shown in Fig. 3a, the PM levels in Beijing have been gradually increasing
from 8:00 to 22:00 on the 26$^{th}$ (Haze II) and from 4:00 to 22:00 on the 22$^{nd}$ (Haze I),
with PM$_{2.5}$ mass concentration eventually reaching 75 μg m$^{-3}$ and 131 μg m$^{-3}$,
respectively. The two stages were respectively regarded as the occurrence stages of
Haze I and Haze II.
*a. The contribution of pollution transport*
Owing to the serious control measures in the summer of Beijing, the sudden
elevated PM levels were very likely to origin the outside region. It's clear that since the
wee hours on the 22$^{nd}$ and 26$^{th}$, Beijing was located behind the northwest-southeast
trough of the 850-hPa potential height field, which bordered the Sichuan Basin to the
west (Fig. 7a-d; Fig. 7i-l). Based on it, Beijing was always under the control of strong
southerly winds at high altitudes. With Taihang mountains to the east and Yanshan
mountains to the north (Fig. 2), Beijing is in semi-enclosed terrain, thus, the south wind
belt passing through the north China plain to Beijing will be strengthened (Su et al.,
2004). The southerly wind speed respectively reached ~8-10 m s$^{-1}$ (Haze II) and ~5-7
m s$^{-1}$ (Haze I) at >0.5 km altitude. As the strong southerly winds persistently blew from
the south, the moisture transport channel, where the water vapor was carried to Beijing
under the southerly winds, was formed and became more and more significant (Fig. 7a-
d; Fig. 7i-l). In response, the humidity in Beijing showed a conspicuous increase in the


morning of 26$^{th}$ with the AH (RH) reaching ~15-17 g m$^{-3}$ (~75 %) while decreased from
10:00 with the AH (RH) down to ~13 g m$^{-3}$ (~70 %) (Fig. 4c-d). The air temperature
during the daytime was extremely high reaching ~30-35 ℃ (Fig. 4b), and high-
temperature weather would reduce the humidity by the evaporation to some degree.
Considering air temperature was always at a distinctly high level (~30 ℃) since the
early morning of 22$^{nd}$, the AH (RH) was ~13 g m$^{-3}$ (~65 %) during the occurrence stage.
With the more populated industrial regions in the south of Beijing, strong winds
blowing from the south were also highly possible to transport heavy anthropogenic
aerosols to Beijing (Chang et al., 2018; Liu et al., 2013b). To explore the potential PM
transportation, we made the distribution maps of PM$_{2.5}$ mass concentration in most parts
of China (Fig. 8) and combined with the corresponding background circulation fields
to try illustrating the pollution transportation. The regional distribution of PM$_{2.5}$ mass
concentration was obtained by interpolating PM$_{2.5}$ data from more than 1000 stations
of China National Environmental Monitoring Centre into grid data (0.5°×0.5°). Noted
that at 2:00 of the 26$^{th}$ and 22$^{nd}$, both the high PM$_{2.5}$ mass concentration (~70 μg m$^{-3}$ for
Haze I and ~50 μg m$^{-3}$ for Haze II) were mainly distributed in the south/southwest of
Beijing, dramatically higher than that (~10 μg m$^{-3}$) in Beijing city (Fig. 8a-b; Fig. 8i-l).
The south area of Beijing heavily polluted was mainly in Baoding, Langfang and
Shijiazhuang and so on, which are generally ~60-300 km far away from Beijing (Fig.
2). The southerly air mass above ~0.5 km moved more than ~20-30 km h$^{-1}$ (estimated
from the measured wind speed) on the 26$^{th}$ and 22$^{nd}$, which is fast enough to transport
pollutants to Beijing in several hours. As expected, the high PM$_{2.5}$ mass concentration
area gradually spread northward sensitively corresponding to the southerly winds, and
consequently, the highest PM$_{2.5}$ level located in Beijing at 20:00 on both the 26$^{th}$
(reaching ~65 μg m$^{-3}$) and 22$^{nd}$ (reaching ~80 μg m$^{-3}$). It was consistent with the rising
trends of PM$_{2.5}$ in this time shown in Fig. 3a. The averaged rising rate of PM$_{2.5}$
concentration (~5.8 μg m$^{-3}$ h$^{-1}$) on the 22$^{nd}$ was higher than that (~3.73 μg m$^{-3}$ h$^{-1}$) on
the 26$^{th}$, possibly related to the bigger difference of PM$_{2.5}$ concentration between
Beijing city and the south area of Beijing. The results were consistent with the findings



reported in Zhong et al. (2018). Thus, multiple results implied that PM transportation
by southerly winds was primarily responsible for the PM rising in the occurrence stage.

### b. The effect of atmospheric boundary layer structure

As shown in Fig. 5a-b, in the forenoon of the 26th and 22nd, the positive values of
the virtual potential temperature gradient ($\partial\theta_v/\partial z$) and pseudo-equivalent potential
temperature gradient ($\partial\theta_{se}/\partial z$) at 0-2 km altitude (Haze II) and at 0-1 km altitude (Haze
I) indicated a stable atmosphere layer existing. Generally, with no solar radiation
reaching the ground and more upward long-wave radiation from the ground in the
nighttime, the surface is cooled faster than the upper atmosphere, promoting a stable
atmosphere. In response, the turbulent kinetic energy (TKE) was extremely low (0-1
$m^2\ s^{-2}$) along with a low ABLH of ~0.5 km (Fig. 5c-d). It means for both the 26th and
22nd, the south winds kept blowing when the ABL structures were not conducive to the
vertical diffusion of substances. The stable ABL structure suppressing the vertical
diffusion of pollution also to some degree contributed to the occurrence of PM pollution.
Both the $\partial\theta_v/\partial z$ and $\partial\theta_{se}/\partial z$ at 0-1.5 km altitude turned to negative at 14:00-16:00 on the
26th, indicating an unstable atmosphere layer. Generally, strong daytime solar radiation
reaching the surface may rebuild the vertical temperature structure and break the stable
ABL, especially in summer. Thus, the turbulence was quickly generated by
thermodynamic activity with TKE growing to ~2-3 $s^2\ m^{-2}$ and continued to develop
upwards causing ABLH gradually increased to ~1.2 km. This ABL process would
explain the slight fluctuations in the PM rising at this time in which the $PM_{10}$ mass
concentration sharply decreased from 100 μg m$^{-3}$ to 73 μg m$^{-3}$. Different from the ABL
condition on the 26th (Haze II), the $\partial\theta_{se}/\partial z$ was negative but the $\partial\theta_v/\partial z$ was positive
below ~1.5 km in the afternoon on the 22nd (Haze I). And combined with the low TKE
(~0-0.5 $m^2\ s^{-2}$) like that in the forenoon, the atmospheric stratification below ~1.5 km
was of absolutely stable state. Maybe little solar radiation heating the ground in the
afternoon on account of cloudy weather, the original stable ABL structure formed at
nighttime cannot be broken. All the above results imply that the ABL structure also
plays a role in the PM rising in the occurrence stage.



***c. The secondary aerosol formation driven by strong atmospheric oxidation***
***capacity***
When the $PM_{2.5}$ increased sensitively to strong southerly winds in Beijing during
the occurrence stage in Haze II (Haze I), $O_3$ showed a sharper growth trend, increasing
rapidly from 67 (26) μg m$^{-3}$ and peaking at 250 (131) μg m$^{-3}$. As mentioned in section
3.1, the high concentration of $O_3$ indicates active atmospheric photochemical reactivity
(Li et al., 2012; Seinfeld, 1986), thus, the atmosphere was of strong oxidizing capacity
with the presence of large amounts of free radicals (OH, etc.) and ozone, which can
promote the formation of secondary aerosols (Pathak et al., 2009; Shi et al., 2015; Wang
et al., 2016). As seen in Fig. 6b, along with the increase in $PM_{2.5}$ concentration in this
occurrence stage, the concentrations of organics, sulfate, and nitrate in $PM_{2.5}$ also
gradually increased. The average concentrations of organics, sulfate, and nitrate in the
occurrence stage of Haze II (Haze I) were 15.6 (23.0) μg m$^{-3}$, 10.0 (8.0) μg m$^{-3}$ and 4.3
(24.7) μg m$^{-3}$ and accounted for 40.7 (32.1) %, 25.3 (11.2) %, and 12.2 (31.5) % to
$PM_{2.5}$ concentration, respectively. The total concentration of sulfate, organics, and
nitrate (SON) accounted for more than 75 % to $PM_{2.5}$ concentration in the occurrence
stage of both Haze II and Haze I (Fig. 6c), implying that the increase of SON is the
leading cause of increase of $PM_{2.5}$ concentration. Secondary organics aerosols can be
formed from the photochemical oxidation reactions of VOCs, emitted by vehicles
(Hennigan et al., 2011). Thus, the high concentration and relative contribution of
organics are mainly attributed to the active photochemical reactions in summer and
huge vehicle emissions of VOCs in Beijing city. Due to the lack of VOCs data, the
detailed formation mechanism of secondary organics would be further studied in the
future. To explore the possible formation mechanism of secondary inorganic aerosols,
sulfur oxidation ratio (SOR) and nitrogen oxidation ratio (NOR) respectively defined
as $SOR = [SO_4^{2-}]/([SO_4^{2-}] + [SO_2])$ and $NOR = [NO_3^-]/([NO_3^-] + [NO_2])$, of which
[ ] stands for the molar concentration were important and used in this paper. Higher
SOR and NOR suggest the higher oxidation efficiency of sulfur and nitrogen, which
means more secondary inorganic aerosols exist in the atmosphere (Liu et al., 2019c;



Han et al., 2019; Yao et al., 2002; Kong et al., 2018; Sun et al., 2006).
Both homogeneous gas-phase and heterogeneous reactions can promote the
formation of sulfate from $SO_2$ during haze episodes (Khoder, 2002; Harris et al., 2013),
increasing the SOR. Noted that SOR values during the whole observation period (from
the 22[nd] to the 27[th]) were relatively high, averaging 0.62, along with relatively low $SO_2$
level, averaging 2.2 μg m$^{-3}$ (Fig. 6a; d). The observed high SOR values could be
attributed to the relatively high RH (averaged ~66.6 %) (Fig. 6e) and ubiquitous
photochemical reactions in summer in Beijing (Han et al., 2019). Nevertheless,
compared to the quite low PM level in clean day (on the 25[th]) (Fig. 6d), the temporal
variation of sulfate concentration on the 26[th] (Haze II) and 22[nd] (Haze I) showed a
distinct increase trend in the occurrence stage, gradually increasing from 3.7 μg m$^{-3}$ to
14.4 μg m$^{-3}$ and from 4.2 μg m$^{-3}$ to 11.5 μg m$^{-3}$, respectively. Meanwhile, SOR values
also at higher level averaged ~0.76 in the occurrence stage of both Haze II and Haze I
than those in clean day averaged ~0.55 (Fig. 6c). The results indicated an enhanced
secondary sulfate aerosol formation in the occurrence stage. However, the PM level and
sulfate concentration in clean day are quite low, but the concentration of $O_3$ is relatively
high (Fig. 6a), reaching up to 214 μg m$^{-3}$, which means active photochemical reactions.
Thus, although the significant photochemical reactions occurred at daytime on the 26[th]
and 22[nd] facilitated the homogeneous gas-phase oxidation of $SO_2$ to a certain extent,
but it is not the dominant reason for the increase of sulfate in the occurrence stage.
Noted that the PM level and total chemical compositions mass slowly increased on the
24[th] with no pollution transportation by south winds (Fig. 3a-b; Fig. 7e-h; Fig. 8e-h;),
while the averaged concentration of sulfate was 2.8 μg m$^{-3}$ and only accounted for
10.7 %, far lower than those in Haze II and similar to those on the clean day. And the
average RH was 61.4 % and 75.3 % in the occurrence stage of Haze II and Haze I,
which also higher than that in the clean day (54.5 %). According to the results
mentioned above, strong winds blowing from the south and southwest of Beijing bring
numerous moisture and particles, we infer that the increase in sulfate aerosols in Haze
II and Haze I could be mainly attributed to the regional transport, then the moisture and



particles transported to Beijing further facilitated the heterogeneous reactions of $SO_2$
on moist aerosol surface. This highlights the importance and urgency of enhancing joint
regional pollution emission control.

Nitrate can be formed predominantly via both the homogeneous gas-phase

photochemical reaction of $NO_2$ with OH radical at daytime when photochemical
activity is high (Wang et al., 2006; Wen et al., 2018; Seinfeld and Pandis, 2006), and
heterogeneous hydrolysis reaction of $NO_3$ and $N_2O_5$ in the atmosphere during the
nighttime (Richards, 1983; Russell et al., 1986; Wang et al., 2009; Wang et al., 2017a;
Pathak et al., 2011). In addition, there is an equilibrium between particulate nitrate and
gaseous $HNO_3$ and $NH_3$ in the atmosphere due to ammonium nitrate is semi-volatile
(Seinfeld, 1986). High temperature could promote the decomposition of ammonium
nitrate, thus, the regional transport of ammonium nitrate in summer was not considered
(Li et al., 2019). As shown in Fig. 6b and d, the nitrate concentration (NOR) in the
occurrence stage of Haze II was lightly increased from 3.2 μg m$^{-3}$ (0.09) at 8:00 to 5.2
μg m$^{-3}$ (0.23) at 22:00. While the nitrate concentration (NOR) in the occurrence stage
of Haze I sharply increased from 2.7 μg m$^{-3}$ (0.02) at 8:00 to 38.1 μg m$^{-3}$ (0.36) at 16:00.
The concentration of nitrate and relative contribution of nitrate to PM during Haze I
were markedly higher than those in Haze II (Fig. 6c). This inconsistency could be
attributed to the high temperature (averaging ~34 ℃) in Haze II than that (averaging
~27 ℃) in Haze I (Fig. 6e). These results indicated that strong photochemical reactions
can facilitate the formation of nitrates, increasing the $PM_{2.5}$ level, while the nitrate
would be decomposed into gaseous $HNO_3$ and $NH_3$ once the temperature is high. After
15:00, the concentration of nitrates began to increase for the presence of large amounts
of radicals and dropped temperature inhibited the reverse reaction. Into the night, the
increase of nitrate aerosols was predominantly through heterogeneous hydrolysis
reaction of $NO_3$ and $N_2O_5$ in the atmosphere, more details would be discussed in the
next section.
**3.2.2 The outbreak stage**

The $PM_{2.5}$ mass concentration suddenly increased from 75 μg m$^{-3}$ at 22:00 on the





26th to 146 μg m$^{-3}$ at 4:00 on the 27th and stayed high values of ~150 μg m$^{-3}$ until 10:00,
which was identified as an outbreak stage of haze pollution (Fig. 3a). Comparing to
the atmospheric BSC of ~2.5-3 M m$^{-1}$sr$^{-1}$ on the 26th, the ambient particles
concentrated below ~0.5 km altitude with a sharply increased atmospheric scattering
coefficient, reaching more than 6 M m$^{-1}$sr$^{-1}$ (Fig. 3b).
### *a. The contribution of southerly transport was almost gone*
There were still strong southerly winds controlling Beijing at high altitude (>0.5
km), accompanied by a more significant vapor transportation channel under it (Fig.
7m-n). However, PM levels in the south/southeast of Beijing, ranging from 0 to ~60
μg m$^{-3}$, were significantly lower than that (>80 μg m$^{-3}$) in Beijing, even below air
quality standards (Fig. 8n-m). It's not likely to make an explosive growth of PM level
and maintain a high PM level in Beijing by pollution transportation.
### *b. Extremely stable ABL structure was the prerequisite for pollution outbreak*
Without the effect of pollution transportation, more attention was paid to the
interior of the local ABL, and Fig. 5 exhibited the temporal variation of the ABL
structure. Both the values of $\partial\theta_v/\partial z$ and $\partial\theta_{se}/\partial z$ turned to positive (~1.5 ℃/100 m and
~2.5 ℃/100 m, respectively) below ~0.3 km altitude, as depicted in Fig. 5a-b. It
implied a very stable lower layer defined as nocturnal stable boundary layer (NSBL)
was formed with ABLH of ~0.3 km. By the strong radiation effect of already-existing
high aerosol loading at daytime, the surface solar radiation could be strongly blocked
and reduced, conducive to a stable stratification formed at midnight (Zhao et al., 2019;
Zhong et al., 2017). In such a thermally stable state, the buoyancy transport heat flux
in the atmosphere will continuously consume turbulent energy, suppressing the
development of turbulence. Therefore, the corresponding TKE was in a sharp decrease
compared to that in 14:00-16:00 on the 26th, lower than ~0.5 m$^2$ s$^{-2}$ even near to ~0 m$^2$
s$^{-2}$ (Fig. 5c-d). However, the values of $\partial\theta_v/\partial z$ and $\partial\theta_{se}/\partial z$ were respectively positive
and negative from ~0.3 km to ~1.5 km, which means this atmospheric layer was of
conditional instability. Considering the quite low TKE like that below ~0.3 km, this
layer recognized as the residual layer was also absolutely stable. Thus, the ambient





particles were restrained from vertically spreading and concentrated below the NSBL,
leading to a growth of the ground PM level. The same work would happen to the
ambient water vapor transported by the southerly winds, which explained the
extremely high humidity during this period. As shown in Fig. 4c-d, the atmospheric
humidity in the outbreak stage was distinctly higher than that on the 26[th] with the AH
(RH) reaching ~20 g m$^{-3}$ (~90 %). Different from the role of moisture transport channel,
the unique NSBL structure has a more significant impact on the increase of air
humidity.
In contrast, for the Haze I on the 22[nd], there was no such thing as an outbreak
stage of PM pollution, as the PM$_{2.5}$ mass concentration had sharply decreased from
131 μg m$^{-3}$ to 53 μg m$^{-3}$ in one hour since 21:00. The reason that the ambient particles
were not accumulated and maintain high level like that in Haze II was the ABL
structure has not met the similar characteristics. The already-existing high PM$_{2.5}$ level
(~130 μg m$^{-3}$) at daytime would accelerate the surface cooling causing the NSBL
formed more easily with a very low height of ~0.2 km. This situation was similar to
that in Haze II. Nevertheless, the TKE above the NSBL was very high reaching ~2-3
m$^2$ s$^{-2}$, in notable contrast to that in Haze II where the TKE was extremely low (~0 m$^2$
s$^{-2}$) in the whole 0-1.5 km layer. The vertical temperature structures above the NSBL
meant the atmosphere was of conditional instability, while in terms of the TKE
distribution, the atmospheric stratification above the NSBL in Haze I was unstable, in
contrast to the stable one in Haze II. Because it was raining at night with the high AH
(~15-20 g m$^{-3}$) and RH (>90 %) extending from surface to ~3 km altitude, the
convection activity was quite strong accompanied by a wet deposition process. Due to
the unstable ABL structure and the accompanying wet deposition, the ambient particle
concentration cannot explosively increase but instead was removed from the
atmosphere.
Noted that the PM on the 24[th] also showed a tendency to increase, but it suddenly
reduced like that in Haze I. There was no transportation effect contributing to the
increase of PM level on the 24[th] with westerly circulation field controlling (Fig. 7e-h).





Similar to the occurrence stages in Haze I and Haze II, a stable atmosphere near the
surface was formed with the positive $\partial \theta_v/\partial z$ and $\partial \theta_{se}/\partial z$. Under these stable
stratifications, the PM from local emission on the 24[th] started increasing. By the strong
daytime solar radiation heating the surface quickly, it may break up the anomalous
vertical temperature structures formed by long-wave radiation cooling at nighttime and
changed them into the unstable stratifications with negative $\partial \theta_{se}/\partial z$ ($\partial \theta_v/\partial z$) profiles. As
discussed in Haze II, the ABL structure characterized by increased TKE ($\sim$2-3 m$^2$ s$^{-2}$)
and elevated ABLH ($\sim$1.5 km) would dissipate the pollution soon. However, the
difference between the Haze II and the pollution process on the 24[th] was that the
unstable atmospheric stratifications with strong TKE on the 24[th] kept developing until
the end of the day, while for Haze II, this condition just lasted two or three hours at
noon. Additionally, an NSBL formed at midnight in Haze II with the ABL height of
$\sim$0.3 km, making the vertical diffusion condition in the near stratum even worse.
Therefore, the subsequent stable atmospheric stratification on the 26[th] was the necessary
premise for the pollution outbreak in Haze II. Particles would not be accumulated and
lead to an outbreak of pollution without a stable ABL structure and can be easily
dissipated by the self-cleaning capacity of the atmosphere.

***c. Intense secondary aerosol formation driven by atmospheric oxidation***

***capacity drove the pollution outbreak***

Heterogeneous aqueous reactions reference to the secondary formation of sulfates

and nitrates largely related to ambient humidity (Wang et al., 2012; Gibson et al., 2007).
The accumulation of water vapor in the NSBL would facility the formation of
secondary aerosols and further driven the outbreak of PM pollution. In order to
investigate the explosive growth mechanisms, we divided the outbreak stage of PM
pollution during Haze II into two stages: Stage I, from 22:00 on the 26[th] to 4:00 on the
27[th]; Stage II, from 5:00 to 10:00 on the 27[th]. During Stage I, along with the explosive
growth of PM$_{2.5}$, the concentration of nitrate rapidly increased from 11.6 µg m$^{-3}$ to 57.8
µg m$^{-3}$, while sulfate and organics sightly increased from 13.7 µg m$^{-3}$ to 19.8 µg m$^{-3}$
and from 21.8 µg m$^{-3}$ to 24.9 µg m$^{-3}$ (Fig. 6d), respectively. During Stage II, the nitrate



concentration stayed the highest value of ~57 μg m$^{-3}$ and the sulfate level maintained
~19 μg m$^{-3}$, with organics slowly dropping (Fig. 6d). The explosive growth trend of
nitrate is the most consistent with that of PM$_{2.5}$. In addition, the average concentrations
of organics, sulfate, and nitrate in the whole outbreak stage were 20.6 μg m$^{-3}$, 15.9 μg
m$^{-3}$ and 43.0 μg m$^{-3}$ and accounted for 22.0 %, 17.8 %, 34.9 %, respectively. Compared
to the occurrence stage, the relative contribution of organics and sulfate to PM$_{2.5}$
decreased significantly, while the contribution of nitrate obviously increased. These
results indicated that the explosive growth of PM$_{2.5}$ concentration was driven by the
sharp increase in nitrate concentration. With strong photochemical reactions at daytime,
the mass concentration of O$_3$ was very high before the outbreak stage, up to 214 μg m$^{-}$
$^3$. NO$_2$ would be produced by O$_3$ reacting with a large amount of NO which was
discharged by vehicle in the evening peak. While NO$_2$ would react with O$_3$ aloft to form
NO$_3$ which will rapidly react with NO$_2$ to form N$_2$O$_5$ at nighttime. During stage I, NOR
rapidly increased from 0.26 to 0.60, which implied the oxidization rate of NO$_2$ sharply
increased in a few hours. Considering NO$_2$ stayed relative low value of ~25 μg m$^{-3}$ and
O$_3$ rapidly decreased from 214 μg m$^{-3}$ to 46 μg m$^{-3}$ in stage I (Fig. 6a), the consumption
process of NO$_2$ was more significant than the generation process. The NO$_2$ produced
by consuming O$_3$ was constantly oxidized by O$_3$ to produce a large amount of N$_2$O$_5$,
resulting in a sharp decline in O$_3$ concentration. Once N$_2$O$_5$ was produced, it would
absorb on the moist particle surfaces and react with water in droplets to form nitrate,
resulting in a sudden increase in nitrate, from 11.6 μg m$^{-3}$ to 57.8 μg m$^{-3}$. During Stage
II, O$_3$ slowly decreased to 34 μg m$^{-3}$ at 6:00 on the 27$^{th}$ and NO$_2$ stayed relatively high
value (~44-51 μg m$^{-3}$), which meant the generation process of NO$_2$ was dominated.
Thus, the oxidization of NO$_2$ was not further increasing with the NOR maintaining
~0.45 during Stage II. Then, the nitrate, formed by the pathway that N$_2$O$_5$ adsorbed on
surfaces and reacts with water in droplets, did not increase anymore, maintaining the
highest mass concentration of ~57 μg m$^{-3}$. The processes mentioned above were
unimportant at daytime because N$_2$O$_5$ was in equilibrium with NO$_3$, that is, NO$_3$ was
photolyzed as well as rapidly destroyed by NO which in turn was present whenever





there were $NO_x$ and sunlight. During both Stage I and Stage II, SOR always maintained
a relatively high level of ~0.95, accompanied by a high RH of ~90 %. High SOR and
RH signified that the heterogeneous reaction dominated the formation of particulate
sulfate during the outbreak stage. The increased amount of sulfate lower than nitrate
may relate to the few emissions of $SO_2$ and massive emission of NO from vast vehicles.
This highlights the importance and urgency of enhancing NOx (vehicles) emission
control.

Contrary to expectations, after the wet deposition process in Haze I, the

concentrations of $PM_{2.5}$, $NO_2$ and the total chemical composition abruptly increased at
0:00 on the 23rd, accompanied by a sharp increase of nitrate and NOR (from 9.3 μg m-
3 to 41.5 μg m-3 and 0.26 to 0.49, respectively). These results may be related to the high
RH (more than 93 %), which facilitated the heterogeneous hydrolysis reaction of $NO_3$
and $N_2O_5$, formed by gas pollutants on NOx and $O_3$ that wet deposition process did not
completely clear.
**3.2.3 The diffusion stage**

Since 10:00 on the 27th, the $PM_{2.5}$ mass concentration had sharply reduced to 50

μg m-3 in three hours, during which the atmospheric BSC down to $<1\times10^3$ M m-1sr-1 on
the whole ABL (Fig.3 and Fig. 8o-p). It represented a stage of pollution diffusion. As
no wet deposition process existed, the diffusion stage of Haze II was different from that
of Haze I. Generally, the arrival of strong and clean air mass from the south is the main
factor that dissipates the air pollution in Beijing (Zhong et al., 2017; Zhong et al., 2018;
Zhao et al., 2019). Calm/light winds in the lower layer were dominated in the outbreak
stage, while sudden increased southerly winds blew in the 0-2 km layer since 8:00 on
the 27th, with a wind speed of ~6-9 m s-1 (Fig. 7n-q and Fig. 4a). Strong winds would
play a role in the horizontal diffusion of the accumulated PM at surface. Then,
accompanied by the horizontal diffusion, the strong solar radiation at noon reached the
surface and changed the vertical temperature structure. The ABL was in extremely
unstable state for both the $\partial\theta_v/\partial z$ and $\partial\theta_{se}/\partial z$ were negative below ~1.0 km with values
of -0.5 °C/100 m and -2.5 °C/100 m, respectively (Fig. 5a-b). Along with the instability,





the development of turbulence in the ABL was very strong and quick, with the TKE
values suddenly increasing to ~3-5 $m^2 s^{-2}$ (Fig. 5c). Accompanied by the pronounced
turbulence development, the ABL continuously developed upward with the ABLH up
to the ~2.5 km in short time (Fig. 5d). The ABL structure quickly became extremely
suitable for the vertical diffusion of pollutants, thus, the PM level sharply decreased
during this time.
Different from $PM_{2.5}$, the concentration of $O_3$ rapidly increased due to the
increasing radiation, along with the high concentrations of $NO_2$ and NO attributed to
morning traffic emissions. Along with the decline in $PM_{2.5}$, organics, and sulfate slowly
decreased to less than ~3 $\mu g\ m^{-3}$ and nitrate reduced to below 1.0 $\mu g\ m^{-3}$. The average
concentrations of organics, sulfate, and nitrate were down to 6.8 $\mu g\ m^{-3}$, 6.2 $\mu g\ m^{-3}$ and
1.9 $\mu g\ m^{-3}$ and accounted for 33.0 %, 32.3 %, 6.0 %, respectively. As significant
turbulence activity made vertical transportation of vapor, heat, and particles and so on,
the RH decreased to ~60 % accompanied by a decline in SOR (~0.75). This emphasized
the strong correlation between humidity and the heterogeneous formation mechanism
of sulfate. In addition, NOR rapidly decreased from 0.22 to 0.01, coincide with the
variation of nitrate. At this stage, temperature always maintained a high level of ~35 ℃.
Thus, similar to the situation in the occurrence stage, ammonium nitrate evaporated
under the high temperature, contributing to a decline in nitrate. In short, during the
diffusion stage, the unstable ABL structure was not only conducive to the diffusion of
pollution but also changed the T and RH, so as to inhibit the formation of secondary
aerosols and further reduce the secondary aerosols.
No matter the wet deposition in Haze I or the vertical diffusion in Haze II,
eventually, the air pollution would be cleared as long as the atmosphere of a specific
state. In other words, it implies that the self-cleaning capacity of the atmosphere is
responsible for the dispersion of air pollution. When the atmosphere is in what state can
the self-cleaning capacity of the atmosphere come into play so that the pollution can be
removed from the atmosphere. To discuss it, the key factors characterizing the self-
cleaning capacity of the atmosphere should be found out first. As analyzed above, once



the TKE increased to >1.5-2 m$^2$ s$^{-2}$, the ABLH grew to more than ~1 km, and the $\partial\theta_v/\partial z$
& $\partial\theta_{se}/\partial z$ turned to negative, as well as no calm/light winds, the atmosphere was in a
state of instability with strong turbulence activities and advection transport, and air
pollution was spread away immediately. Owing to limited observation time, the results
about the characteristics of self-cleaning capacity of the atmosphere may be not
universal, and more comprehensive discussions on the self-cleaning capacity of the
atmosphere would be studied in the future.

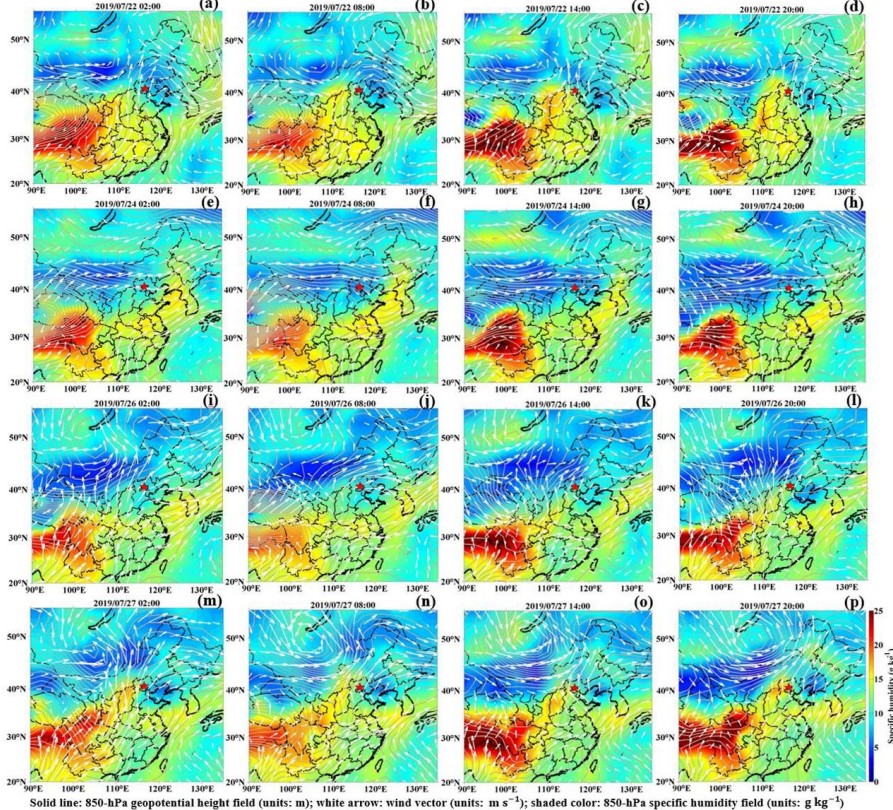

Solid line: 850-hPa geopotential height field (units: m); white arrow: wind vector (units: m s$^{-1}$); shaded color: 850-hPa specific humidity field (units: g kg$^{-1}$)
Figure 7. Composites of the 850-hPa horizontal wind vector field (units: m s$^{-1}$, white arrows),
850-hPa geopotential height field (units: m, solid lines) and 850-hPa specific humidity field
(units: g kg$^{-1}$, shaded colors) at 0200, 0800, 1400, and 2000 (LT) on July 22, 24 and 26-27,
labeled as (a) - (p). The star shows the location of the BJ site.

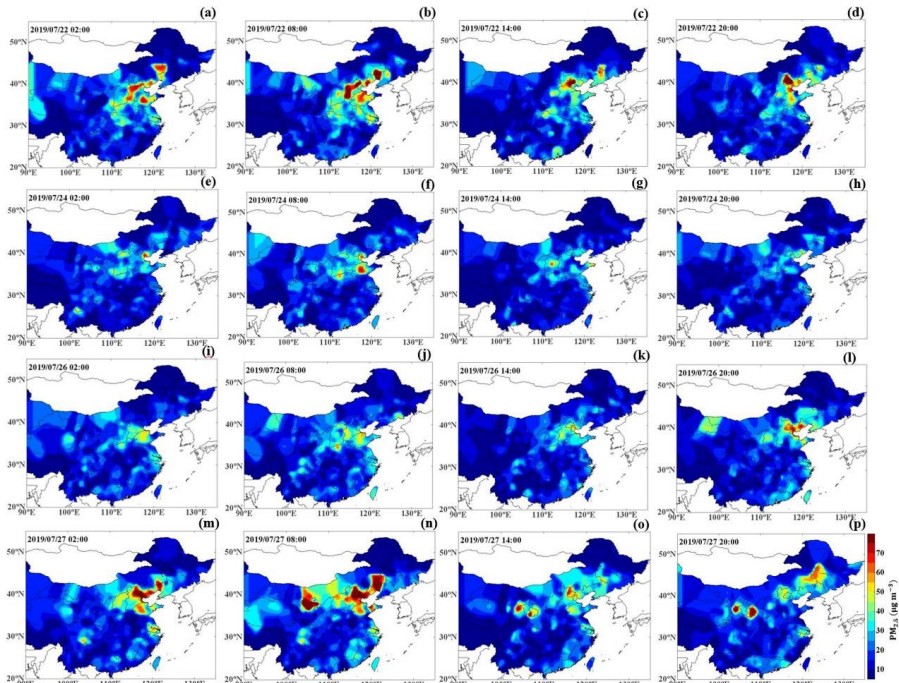


Figure 8. The PM$_{2.5}$ mass concentration distribution (units: μg m$^{-3}$, shaded colors) over most of

China at 0200, 0800, 1400, and 2000 (LT) on July 22, 24 and 26–27, labeled as (a)–(p).

## 4    Conclusion

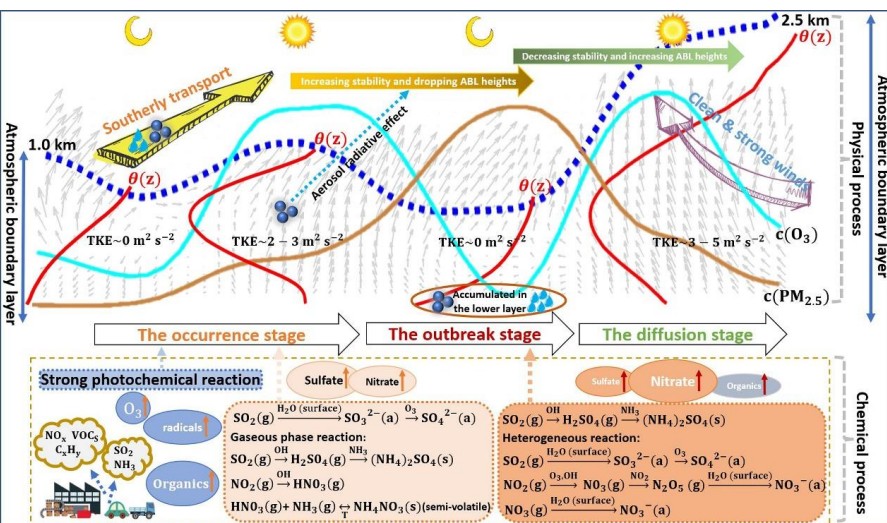


Figure 9. A schematic diagram for the formation mechanism of haze pollution under strong

atmospheric oxidization capacity in summer in Beijing.



The extremely serious haze pollution episode characterized by
alternate/synchronous heavy PM loading and high ozone concentration occurred this
summer in Beijing. Combined with a series of observations, the formation mechanism
of haze pollution under strong atmospheric oxidization capacity has been deeply and
systematically analyzed in terms of atmospheric physical process and chemical process.
The occurrence of haze pollution in summer in Beijing was mainly attributed to
southerly transport and affected by the ABL structure to some degree (physical process),
which was further promoted by the intense secondary aerosol formation with strong
atmospheric oxidation capacity (chemical process). On the one hand, the physical
process, where large amounts of moisture and particles were transported to Beijing
under strong southerly winds, caused the initiation of haze pollution in Beijing.
Moreover, it occurred when the ABL structure was extremely stable with low TKE and
positive potential temperature gradient ($\partial\theta/\partial z$), facilitating the PM level rising in
Beijing. And the stable ABL was broken and transformed into unstable (negative $\partial\theta/\partial z$)
with strong solar radiation in the afternoon, responsible for the fluctuation of the PM
rising process. On the other hand, the moisture and particles transported to Beijing
further facilitated the heterogeneous reactions of $SO_2$ on moist aerosol surfaces. And
for the significant photochemical reaction, the concentration of $O_3$ was quite high at
daytime and the atmosphere was of strong oxidation capacity with large amounts of
radicals (OH, etc.) and $O_3$, promoting the formation of secondary aerosols (sulfate,
nitrate, and organics). Even so, the distinct increase in sulfate concentration was mainly
linked by southerly transport, which carried heavy sulfate aerosol loading to Beijing.
The physical process, where extremely stable ABL inhibited the diffusion of PM and
moisture making an accumulation of ambient humidity and ground-level $PM_{2.5}$, was the
premise of the outbreak of haze pollution. Under stable ABL, the formation of
secondary aerosols dominated by nitrate was quite intense, and this pronounced
chemical process was the key driving force leading to pollution outbreak. PM levels in
the south/southeast of Beijing were significantly lower than that in Beijing, even below
air quality standards. The contribution of pollution transportation was not important.



Owing to the already-existing high $PM_{2.5}$ level at daytime, strong aerosol radiation
effect would cool the surface and heat the above layer, facilitated the formation of the
nocturnal stable boundary layer (NSBL). The $\partial\theta/\partial z$ in the NSBL turned to positive
increasing the atmosphere stability, dropping the ABLH and decreasing the TKE. The
ambient particles & moisture were restrained from vertically spreading and
concentrated below the NSBL, resulting in an elevated PM & humidity levels at surface.
Due to the high level of $O_3$ produced by strong photochemical reactions at daytime and
NOx discharged by vehicle in the evening peak, vast $N_2O_5$ and $NO_3$ were formed with
a sharp increase of NOR. The heterogeneous hydrolysis reactions of $N_2O_5$ and $NO_3$ at
the moist particle surface were very significant under quite high humidity. It resulted
in the formation of numerous nitrate, which was the main cause of the explosive growth
of $PM_{2.5}$ levels in the outbreak stage. More controls should be made to reduce NOx
emissions and atmospheric oxidization capacity, such as strengthening the supervision
of heavy diesel vehicles and collaborative control of NOx and VOCs, and continuously
deepen regional joint control of air pollution. As PM level gradually increasing, a
wet deposition process and an extremely unstable ABL structure respectively appeared
on the $22^{nd}$ (Haze I) and $24^{th}$, the ambient particles experienced a sharp decline before
the outbreak stage. It emphasized that the ABL structure extremely restrained the
diffusion of substances was a prerequisite for the pollution outbreak. With clean &
strong winds passing through Beijing, the ABL changed to unstable with negative $\partial\theta/\partial z$
and increased ABLH. The strong turbulence activity promoted pollution diffusion. No
matter the wet deposition process or the strong turbulence activity, eventually, the air
pollution would be cleared as long as the atmosphere was of a specific state. The self-
cleaning capacity of the atmosphere is responsible for the dispersion of air pollution.
When the atmosphere is in what state can the self-cleaning capacity of the atmosphere
comes into play is worthy of further study.
**Data availability.**
The surface $PM_{2.5}$ & $PM_{10}$ and other trace gases observation data used in this study can
be accessed from http://106.37.208.233:20035/. Atmospheric reanalysis data was



obtained from the National Centers for Environmental Prediction (NCEP)
(https://www.esrl.noaa.gov/psd/data/). Other datasets can be accessed upon request to
the corresponding author.

**Author contribution**


ZD and LG performed the research and wrote the paper, contributing equally to this
study. XJ, QJ, WY and WX provided writing guidance, revised and polished the paper.
LZ, TG, HB and WL designed the experiments and DL, MY, WX and WF carried them
out. GC contributed to discussions of results. All the authors have made substantial
contributions to the work reported in the manuscript.

**Competing interests.**


The authors declare that they have no conflict of interest.

**Acknowledgments**


This study was supported by the Ministry of Science and Technology of China
(grant number 2016YFC0202001), the CAS Strategic Priority Research Program
(XDA23020301) and the National Natural Science Foundation of China (grant number
41375036). The authors are grateful for services rendered by the National Oceanic and
Atmospheric Administration (NOAA) and National Centers for Environmental
Prediction (NCEP). The authors are thankful for the data support from the National
Earth System Science Data Sharing Infrastructure, National Science & Technology
Infrastructure of China (available at http://www.geodata.cn).

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
