# Peer review of "Haze pollution under a high atmospheric oxidization capacity in summer in"

_Atmospheric Chemistry and Physics, 2019_

## Referee Comment (RC1) · Anonymous Referee #3 · 21 Jan 2020

This study of haze pollution during summer in Beijing, as shown in this paper, provides the synergistic effect of physico-chemical processes in the atmospheric boundary layer (ABL). The seriously polluted surrounding areas of Beijing in the South/Southwest are generally 60 up to 300 km away from Beijing. Beijing can be in contrast clean. Southerly winds of more than 20 up to 30 km h-1 during early morning transport these pollutants to Beijing and initiate primarily the haze formation. During daytime the PM2.5 level increases to 75 $\mu$g m-3 during several hours in Beijing, supported by a simultaneously stable ABL structure. Additionally, the O3 concentration is quite high at daytime

(250 $\mu$g m-3), showing a strong atmospheric oxidation capacity. Significant sulfate and nitrate concentrations are formed through atmospheric chemical processes with a sulfur oxidation ratio (SOR) up to 0.76 and a nitrogen oxidation ratio (NOR) which increases from 0.09 to 0.26 so that the particulate matter (PM) concentration level is rising further. Even so, the increase in sulfate is mainly linked to southerly transport. During night the PM2.5 concentration is sharply increasing from 75 $\mu$g m-3 up to 150 $\mu$g m-3 during 4 hours and persists at that high level until the next morning. With simultaneous extremely stable ABL structure the formation of secondary aerosols which is dominated by nitrate is quite intense, so that this configuration is driving the outbreak of a haze pollution. In that case the PM concentration levels in the South/Southeast of Beijing are significantly lower than in Beijing, even below air quality standards, because the contribution of pollution transport is almost neglectable. Corresponding to the formation of a nocturnal stable boundary layer height at lower than 0.3 km, the extremely low turbulence kinetic energy (TKE) of up to 0.05 m2 s-2 inhibits the spread of particles and moisture and causes elevated levels of PM2.5 and relative humidity (about 90 %) near to the surface. Under quite high humidity and strong ambient oxidization capacity, the NOR rapidly increases from 0.26 to 0.60 and heterogeneous hydrolysis reactions at the wet particle surface are very significant. The nitrate concentration explosively increases from 11.6 $\mu$g m-3 up to 57.8 $\mu$g m-3, while the concentrations of sulfate and organics slightly increase by 6.1 $\mu$g m-3 and 3.1 $\mu$g m-3, respectively. With clean and strong winds passing over Beijing, the stable ABL is broken with potential temperature gradients turning to negative values and ABL heights increasing to about 2.5 km. The strong turbulence activity caused by TKE values of 3 up to 5 m2 s-2 notably supports the pollution diffusion. So, the self-cleaning capacity of the atmosphere is always responsible for the dispersion of air pollution. Even so, the reduction of atmospheric oxidization capacity by strengthening the collaborative decrease of nitrogen oxide (NOx) and volatile organic compounds (VOCs) emissions is urgent as well as the continuous regional joint decrease of air pollutant emissions. General comments This study shows the complexity of haze formation processes due to coupling of transport, turbulence,

stability of the lower atmosphere as well as chemical reaction. The central rule of atmospheric oxidization capacity is found and their influence upon the haze generation is described. The new results of this study are well described and discussed in relation to the state-of-the-art research of haze formation. It would be helpful for the whole understanding of haze in Beijing if the influence of haze upon the radiation transfer in the atmosphere and thus the transport, turbulence and stability, which is mentioned in the introduction, or one can say the self-cleaning capacity of the atmosphere (that means feedback mechanisms) is described also. It is shown in Fig. 9 only. The conclusions are a summary only and in this summary no relations to the existing knowledge / papers are given. Thus, a discussion of the results in relation of the state-of-the-art knowledge about summer haze in Beijing is required so that one can follow what is new and what is supported by this study. The conclusions in the last sentence of the abstract must be given and discussed in the chapter conclusions. The paper addresses relevant scientific questions within the scope of ACP. The paper presents novel concepts, ideas, tools and data. The scientific methods and assumptions are valid and clearly outlined so that substantial conclusions are reached. The description of experiments and calculations allow their reproduction by fellow scientists. The results are sufficient to support the interpretations and conclusions. The quality of the figures is good. The figure captions should be improved so that these are understandable without the overall manuscript: terms must be explained, description of parameters. The related work is well cited so that the authors give proper credit to related work and own new contribution. The title reflects the whole content of the paper. The abstract provides a concise and complete summary. The overall presentation is well structured and clear. The language is fluent and precise but must be improved in very much details. It is necessary that a native speaker is improving the manuscript. The mathematical formulae, symbols, abbreviations, and units are generally correctly defined and used. No parts of the paper (text, formulae, figures, tables) should be reduced, combined, or eliminated. The number and quality of references is appropriate. Specific Comments Figure caption 1: Set the letters a) – f) to the single instruments. Line 581: Why diffusion stage if wind increased

(line 589)? Use Dispersion stage? Technical corrections Line 727, 873: doi number is missing. Lines 770, 772: the reference is incomplete. Lines 856, 858, 859: improve the format.

---

## Referee Comment (RC2) · Anonymous Referee #1 · 14 Feb 2020

This paper focusses in detail on the physical and chemical processes involved in 2 extreme summer haze events in Beijing. In particular the paper looks into the coupling between the build of ozone and the resultant feedbacks on PM concentrations through the impacts on the oxidative capacity of the atmosphere. Overall, the paper also concludes the strong role that regional transport plays in such events. Finally the paper also concludes that meteorology plays an important role in the self cleaning effect of the atmosphere that ends such pollution events. Overall this paper is well written and uses robust methodology to support the above conclusions. However there

are some relatively minor corrections that need addressing before publication can be recommended. In particular, there were a number of points in the paper where sentences where repeated. A number of these have been pointed out below but please ensure all are corrected in the revised manuscript.

Minor comments: Page 1, Line 18: Please remove the $\sim$ symbol at the end of the line. No need to put approximations when quoting a range. Page 10, line 232: No need to reference GB3095-2012 again as you have already done this above. Page 15, Line 294: Please correct origin to originate. Page 15, Line 295: Please correct wee hours to early hours Page 15, Line 303: There is a lot of repetition in this sentence. Please simplify to 'Under persistent southerly winds, water vapour was carried to Beijing forming a moisture transport channel which strengthened.' Page 15, Line 305: Please remove conspicuous and just say increase. Page 22, Line 481: I don't understand why the phrasing 'The same work . . ..' Is used here. Please define.
* * *

---

## Author Comment (AC1) · 16 Feb 2020

**General Comments**

This study shows the complexity of haze formation processes due to coupling of transport, turbulence, stability of the lower atmosphere as well as chemical reaction. The central rule of atmospheric oxidization capacity is found and their influence upon the haze generation is described. The new results of this study are well described and discussed in relation to the state-of-the-art research of haze formation. It would be helpful for the whole understanding of haze in Beijing if the influence of haze upon the radiation transfer in the atmosphere and thus the transport, turbulence and stability, which is mentioned in the introduction, or one can say the self-cleaning capacity of the atmosphere (that means feedback mechanisms) is described also. It is shown in Fig. 9 only. The conclusions are a summary only and in this summary no relations to the existing knowledge / papers are given. Thus, a discussion of the results in relation of the state-of-the-art knowledge about summer haze in Beijing is required so that one can follow what is new and what is supported by this study. The conclusions in the last sentence of the abstract must be given and discussed in the chapter conclusions. The paper addresses relevant scientific questions within the scope of ACP. The paper presents novel concepts, ideas, tools and data. The scientific methods and assumptions are valid and clearly outlined so that substantial conclusions are reached. The description of experiments and calculations allow their reproduction by fellow scientists. The results are sufficient to support the interpretations and conclusions. The quality of the figures is good. The figure captions should be improved so that these are understandable without the overall manuscript: terms must be explained, description of parameters. The related work is well cited so that the authors give proper credit to related work and own new contribution. The title reflects the whole content of the paper. The abstract provides a concise and complete summary. The overall presentation is well

structured and clear. The language is fluent and precise but must be improved in very much details. It is necessary that a native speaker is improving the manuscript. The mathematical formulae, symbols, abbreviations, and units are generally correctly defined and used. No parts of the paper (text, formulae, figures, tables) should be reduced, combined, or eliminated. The number and quality of references is appropriate.

Response: Thank the reviewer for the constructive comments and suggestions. According to the reviewer's suggestions, we have done our best to revise our manuscript. The modifications have been highlighted in yellow or red in the revised manuscript.

1. The central rule of atmospheric oxidization capacity is found and their influence upon the haze generation is described. The new results of this study are well described and discussed in relation to the state-of-the-art research of haze formation. It would be helpful for the whole understanding of haze in Beijing if the influence of haze upon the radiation transfer in the atmosphere and thus the transport, turbulence and stability, which is mentioned in the introduction, or one can say the self-cleaning capacity of the atmosphere (that means feedback mechanisms) is described also. It is shown in Fig. 9 only.

Response: Thank the reviewer for the comments and suggestions. About this suggestion, we'd like to address that the figure 9 is an abstract graph which concluded the whole results in this paper and showed the central topic as a schematic diagram. In Fig. 9, the influence of haze upon the radiation transfer in the atmosphere and thus the transport, turbulence and stability, as well as the self-cleaning capacity of the atmosphere were schematically shown, which has been deeply discussed in section 3.2. During each stage of haze episodes, we analyzed the potential causes based on the physical and chemical processes. The physical processes involved the pollution transport and boundary layer structure effect. And the radiation transfer, stability and turbulence have been combined

to discuss the variation of boundary layer structure and its effects on the haze formation. For example, in the occurrence stage, we discussed the pollution causes in terms of transport, aerosol-radiation effect on stability and thus the turbulent activity in the boundary layer and further chemical processes, which was respectively shown in section 3.2.1a, 3.2.1b, and 3.2.1c. The section 3.2.3 discussed the haze diffusion stage and proposed the concept of self-cleaning capacity of the atmosphere.

2. The conclusions are a summary only and in this summary no relations to the existing knowledge / papers are given. Thus, a discussion of the results in relation of the state-of-the-art knowledge about summer haze in Beijing is required so that one can follow what is new and what is supported by this study.

Response: Thank the reviewer for the constructive suggestions. As you suggested, we have added a detailed discussion of the results in relation of the state-of-the-art knowledge about summer haze in Beijing in the chapter Conclusions. The modifications have been highlighted in yellow in the revised manuscript.

3. The conclusions in the last sentence of the abstract must be given and discussed in the chapter conclusions.

Response: Thank the reviewer for the constructive suggestions. In the last sentence of the abstract, we propose that "Even so, reducing atmospheric oxidization capacity such as strengthening the collaborative control of nitrogen oxide (NOx) and volatile organic compounds (VOCs) was urgent, as well as continuously deepening regional joint control of air pollution". As you suggested, we have added a more detailed discussion on it in the chapter Conclusions. The modifications have been highlighted in yellow in the revised manuscript.

4. The figure captions should be improved so that these are understandable without the overall manuscript: terms must be explained, description of parameters.

Response: Thank the reviewer for the constructive suggestions.

Figure caption 2 has been corrected to "Figure 2. Scatter plot of the relationship between the directly measured $PM_{2.5}$ mass concentration (with the particulate matter analyzer of the China National Environmental Monitoring Center) and the ACSM-measured $PM_{2.5}$ mass concentration (the sum of the chemical constituent mass concentrations measured with the aerosol chemical speciation monitor (ACSM) and the black carbon (BC) mass concentration measured with the multiangle absorption photometer).".

Figure caption 7 has been corrected to "Figure 7. Composites of the 850-hPa horizontal wind vector field (units: m s$^{-1}$; white arrows), 850-hPa geopotential height field (units: m; solid lines) and 850-hPa specific humidity field (units: g kg$^{-1}$; shaded colors) at 0200, 0800, 1400, and 2000 (local time) on 22 and 24 July and from 26-27 July, labeled as (a) - (p). The star shows the location of the BJ site.".

Figure caption 8 has been corrected to "Figure 8. The $PM_{2.5}$ mass concentration distribution (units: μg m$^{-3}$; shaded colors) over most of China at 0200, 0800, 1400, and 2000 (local time) on 22 and 24 July and from 26–27 July, labeled as (a)–(p).".

Figure caption 9 has been changed to "Figure 9. Schematic diagram for the formation mechanism of haze pollution under a high atmospheric oxidization capacity in summer in Beijing (blue dashed line: atmospheric boundary layer; red solid lines: potential temperature gradient profiles; brown solid line: temporal change curve of the ozone concentration; cyan solid line: temporal change curve of the $PM_{2.5}$ mass concentration; gray arrow sectors: temporal change in the wind vector profiles; TKE: turbulence kinetic energy; solid dots: particulate matter in the atmosphere; droplets: water vapor)."

5. The language is fluent and precise but must be improved in very much details. It is necessary that a native speaker is improving the manuscript.

Response: Thank the reviewer for the constructive suggestions. Regarding the language, we accepted the suggestion and the revised manuscript has been improved by a native

speaker. The modifications have been highlighted in red in the revised manuscript.

**Specific Comments**

1.  Figure caption 1: Set the letters a) – f) to the single instruments.

Response: Thank you for the suggestion.   The sentence "The pictures of (a)-(f) are Microwave Radiometer, 3D Doppler Wind Lidar, CIMEL sun-photometer, Ceilometer, Aerodyne Aerosol Chemical Speciation Monitor and Multi-angle Absorption Photometer set in the BJ site." in Figure caption 1 has been corrected to "(a: microwave radiometer; b: 3D Doppler wind lidar; c: CIMEL sun-photometer; d: ceilometer; e: Aerodyne aerosol chemical speciation monitor (ACSM); f: multiangle absorption photometer).".

2.  Line 581: Why diffusion stage if wind increased (line 589)? Use Dispersion stage?

Response: Thanks for the comments and suggestions. Regarding the statement in line 589, there is some explanation we need to make. As discussed in this study, the initialization of haze in Beijing was mainly attributed to the southwest/south winds which came through the heavy polluted areas. However, the strong and air mass dissipating the air pollution in Beijing mainly came from the southeast of Beijing, shown in Figure 7(n)-(p). The southeast winds originated from the Bohai Sea and the Yellow Sea. Moreover, during this diffusion stage, the air quality of the southeast of Beijing was basically clean or much better than that in Beijing (Figure 8(n)-(p)). Therefore, strong southeast winds would not bring pollutants aggravating the pollution in Beijing instead played a role in the horizontal diffusion of the accumulated PM at the surface. On the other hand, accompanied by the horizontal diffusion, the strong solar radiation at noon reached the surface and changed the vertical temperature structure. The ABL was in extremely unstable state for both the $\partial\theta v/\partial z$ and $\partial\theta se/\partial z$ were negative below ~1.0 km with values of -0.5 ℃/100 m and -2.5 ℃/100 m, respectively (Fig. 5a-b). Along with this instability, the development of turbulence in the ABL was very

strong and quick, with the TKE values suddenly increasing to ~3-5 m$^2$ s$^{-2}$ (Fig. 5c). Accompanied by the pronounced turbulence development, the ABL continuously developed upward with the ABLH up to the ~2.5 km over short time (Fig. 5d). The ABL structure quickly became extremely suitable for the vertical diffusion of pollutants, thus, the PM level sharply decreased during this time. We may haven't state it more clearly and thus we have supplemented a more detailed discussion on it. The modifications have been highlighted in yellow in the revised manuscript. Through our discussion, we think "Diffusion stage" is appropriate and also the "Dispersion stage".

3. Technical corrections Line 727, 873: doi number is missing. Lines 770, 772: the reference is incomplete. Lines 856, 858, 859: improve the format.

Response: Thanks for the reviewer's suggestions.

The reference in Line 727 has been corrected to "Ainsworth, E. A., Yendrek, C. R., Sitch, S., Collins, W. J., and Emberson, L. D.: The Effects of Tropospheric Ozone on Net Primary Productivity and Implications for Climate Change, Annual Review of Plant Biology, 63, 637-661, https://doi.org/10.1146/annurev-arplant-042110-103829, 2012.".

The reference in Line 873 has been corrected to "Su, F., Gao, Q., Zhang, Z., Ren, Z., and Yang, X.: Transport Pathways of Pollutants from Outside in Atmosphere Boundary Layer, Res. Environ. Sci., 17(1), 26-29,40, https://doi.org/10.3321/j.issn:1001-6929.2004.01.005, 2004.".

Through our discussion, we have decided to delete the reference in Line 770.

The reference in Line 772 has been corrected to "Gregory, L.: Cimel Sunphotometer (CSPHOT) Handbook, Office of Scientific & Technical Information Technical Reports, https://doi.org/10.2172/1020262, 2011.".

The reference in Lines 856-857 has been corrected to "Richards, L. W.: comments on the oxidation of NO2 to nitrate- day and night, Atmos. Environ., 17, 397-402, https://doi.org/10.1016/0004-6981(83)90057-4, 1983.".

The reference in Lines 858-859 has been corrected to "Russell, A. G., Cass, G. R., and Seinfeld, J. H.: On some aspects of nighttime atmospheric chemistry, Environ. Sci. Technol., 20, 1167-1172, https://doi.org/10.1021/es00153a013, 1986.".

**Haze pollution under a high atmospheric oxidization capacity in summer in Beijing: Insights into formation mechanism of atmospheric physicochemical processes**

Dandan Zhao[†1,2]; Guangjing Liu[†3,1]; Jinyuan Xin[*1,2,4]; Jiannong Quan[5]; Yuesi Wang[1]; Xin Wang[3]; Lindong Dai[1]; Wenkang Gao[1]; Guiqian Tang[1]; Bo Hu[1]; Yongxiang Ma[1]; Xiaoyan Wu[1]; Lili Wang[1]; Zirui Liu[1]; Fangkun Wu[1]

1 State Key Laboratory of Atmospheric Boundary Layer Physics and Atmospheric Chemistry (LAPC), Institute of Atmospheric Physics, Chinese Academy of Sciences, Beijing 100029, China

2 University of Chinese Academy of Sciences, Beijing 100049, China

3 College of Atmospheric Sciences, Lanzhou University, Lanzhou 730000, China.

4 Collaborative Innovation Center on Forecast and Evaluation of Meteorological Disasters, Nanjing University of Information Science and Technology, Nanjing 210044

5 Institute of Urban Meteorology, Chinese Meteorological Administration, Beijing, China

(†) These authors contributed equally to this study.

(*) Correspondence: Jinyuan Xin (xjy@mail.iap.ac.cn)

**Abstract:** Under a high atmospheric oxidization capacity, the synergistic effect of the physicochemical processes in the atmospheric boundary layer (ABL) caused summer haze pollution in Beijing. The south/southwest areas, generally ~60-300 km away from Beijing, were seriously polluted, in contrast to Beijing, which remained clean. Southerly winds moving faster than ~20-30 km h$^{-1}$ since the early morning primarily caused haze pollution initiation. The PM$_{2.5}$ (particulate matter with a dynamic equivalent diameter smaller than 2.5 μm) level increased to 75 μg m$^{-3}$ over several hours during the daytime, which was simultaneously affected by the ABL structure. Additionally, the O$_3$ concentration was quite high during the daytime (250 μg m$^{-3}$), corresponding to a high atmospheric oxidation capacity. Much sulfate and nitrate were produced through active atmospheric chemical processes, with sulfur oxidation ratios (SORs) up to ~0.76 and nitrogen oxidation ratios (NORs) increasing from 0.09 to 0.26, which further facilitated particulate matter (PM) level enhancement. However, the increase in sulfate was mainly linked to southerly transport. At midnight, the PM$_{2.5}$ concentration sharply increased from 75 to 150 μg m$^{-3}$ over 4 hours and remained at its

highest level until the next morning. Under an extremely stable ABL structure, secondary aerosol formation dominated by nitrate was quite intense, driving the haze pollution outbreak. The PM levels in the south/southeast area of Beijing were significantly lower than those in Beijing at this time, even below air quality standards; thus, the contribution of pollution transport had almost disappeared. With the formation of a nocturnal stable boundary layer (NSBL) at an altitude ranging from 0-0.3 km, the extremely low turbulence kinetic energy (TKE) ranging from 0-0.05 $m^2$ $s^{-2}$ inhibited the spread of particles and moisture, ultimately resulting in elevated near-surface $PM_{2.5}$ and relative humidity (~90%) levels. Due to the very high humidity and ambient oxidization capacity, NOR rapidly increased from 0.26 to 0.60, and heterogeneous hydrolysis reactions at the moist particle surface were very notable. The nitrate concentration steeply increased from 11.6 to 57.8 μg $m^{-3}$, while the sulfate and organics concentrations slightly increased by 6.1 and 3.1 μg $m^{-3}$, respectively. With clean and strong winds passing through Beijing, the stable ABL dissipated with the potential temperature gradient becoming negative and the ABL height (ABLH) increasing to ~2.5 km. The high turbulence activity with a TKE ranging from ~3-5 $m^2$ $s^{-2}$ notably promoted pollution diffusion. The self-cleaning capacity of the atmosphere is commonly responsible for air pollution dispersion. However, reducing the atmospheric oxidization capacity, through strengthening collaborative control of nitrogen oxide (NOx) and volatile organic compounds (VOCs), is urgent, as well as continuously deepening regional joint air pollution control.

**1  Introduction**

Due to a series of stringent emission control measures (China's State Council 2013 Action Plan for Air Pollution Prevention and Control available at http://gov.cn/zwgk/2013-09/12/), including shutting down heavily polluting factories and replacing coal with clean energy sources, the annual mean $PM_{2.5}$ (particulate matter with a dynamic equivalent diameter smaller than 2.5 μm) concentration in major regions, especially in Beijing, has continuously decreased in recent years (Chen et al., 2019; Liu et al., 2019a; Cheng et al., 2019a; Ding et al., 2019). However, the ground-level $O_3$ concentration across China has increased rapidly in recent years, especially in

summer, despite recent reductions in $SO_2$ and nitrogen oxide (NOx) emissions (Chen et al., 2018; Anger et al., 2016; Wang et al., 2018; Wang et al., 2017b). This discrepancy in the variation trend between $O_3$ and $PM_{2.5}$ may be attributed to inappropriate reduction ratios of NOx and volatile organic compounds (VOCs) in $PM_{2.5}$-control-oriented emission reduction measures, which mainly focus on NOx reduction (Liu et al., 2013a; Cheng et al., 2019b). In addition, a number of studies have demonstrated that reducing ambient particles influences surface ozone generation by affecting heterogeneous reactions and decreasing the photodecomposition rate ($O_3$ and its precursors) through aerosol-radiation interactions (Liu et al., 2019b; Wang et al., 2019b; He and Carmichael, 1999; Dickerson et al., 1997; Tie et al., 2001; Martin et al., 2003; Tie et al., 2005). Recently, even though the $PM_{2.5}$ level in Beijing has generally been low due to stringent emission control measures, several haze pollution episodes with alternating/synchronous high ozone concentrations have still occurred in the summer of 2019. Regarding the causes of particulate matter (PM) pollution, numerous previous studies have reported that stationary synoptic conditions, local emissions and regional transport, an adverse atmospheric boundary layer (ABL) structure and meteorological conditions as well as secondary aerosol formation are major factors in haze pollution formation (Li et al., 2019; Sun et al., 2012; Wang et al., 2016; Liu et al., 2019c; Huang et al., 2017; Luan et al., 2018; Han et al., 2019). Huang et al. (2017) demonstrated that haze pollution in Beijing-Tianjin-Hebei usually occurred when air masses originating from polluted industrial regions in the south prevailed and are characterized by high $PM_{2.5}$ loadings with considerable contributions from secondary aerosols. Bi et al. (2017) stated that the strong winds and vertical mixing in the daytime scavenged pollution, and the weak winds and stable inversion layer in the nighttime promoted air pollutant accumulation near the surface. Zhong et al. (2018) showed that positive ABL meteorological feedback on the $PM_{2.5}$ mass concentration explains over 70% of the outbreak of pollution. Zhao et al. (2019) also revealed that the constant feedback effect between aerosol radiative forcing and ABL stability continually reduced the atmospheric environmental capacity and aggravated air pollution. The dominant PM components, including sulfate, nitrate, ammonium, and organics (Org), are mostly

formed via the homogeneous/heterogeneous reactions of gas-phase precursors in the atmosphere (Orrling et al., 2011; Wang et al., 2016) and account for over 50% of the $PM_{2.5}$ mass (Wang et al., 2013; Liu et al., 2019a; Sun et al., 2015; Yao et al., 2002). Ming et al. (2017) proved that the contribution of secondary aerosol formation during haze pollution episodes was much higher than that before and after haze pollution episodes.

Although the causes of high $PM_{2.5}$ loadings have been widely examined, most of these studies have focused on haze pollution in winter and only involved one or several key factors. In summer in Beijing, with high solar radiation, $O_3$ can be quickly formed via photochemical reactions among precursors, including volatile organic compounds (VOCs) and nitrogen oxides (NOx), which contributes to an increase in the ambient oxidizing capacity (Wang et al., 2017c; Ainsworth et al., 2012; Hassan et al., 2013; Trainer et al., 2000; Sillman, 1999). Meteorological conditions, including solar radiation, temperature, relative humidity, wind speed and direction, and cloud cover, also play an important role in short-term ozone variations, further affecting the atmospheric oxidization capacity (Lu et al., 2019; Cheng et al., 2019b; Toh et al., 2013; Wang et al., 2017d; Zeng et al., 2018). As ozone pollution is increasingly becoming prominent and the atmospheric oxidation capacity is gradually increasing, the formation mechanism of haze pollution under a high atmospheric oxidization capacity needs to be concerned. Previous studies have demonstrated that intense atmospheric photochemical reactions in summer enhanced secondary aerosol formation and led to the synchronous occurrence of high $PM_{2.5}$ and $O_3$ concentrations on a regional scale (Pathak et al., 2009; Wang et al., 2016; Shi et al., 2015). Nevertheless, the mechanisms of how the overall regional transport, ABL structure, meteorological conditions and secondary aerosol formation interact to quantitatively influence haze pollution under a high atmospheric oxidization capacity in summer remain unclear. Therefore, by closely monitoring air temperature and relative and absolute humidity profiles, vertical velocity and horizontal wind vector profiles, atmospheric backscattering coefficient (BSC) profiles and the ABL height (ABLH), as well as the mass concentration and composition of $PM_{2.5}$, aerosol optical depth (AOD) and mass concentrations of gas

pollutants including O₃, SO₂, and NO₂, this paper comprehensively examines the formation mechanism of haze pollution under a high ambient oxidization capacity insights into atmospheric physics and chemistry to propose select recommendations for model forecasting and cause analysis of complex air pollution in summer in Beijing.

**2 Instruments and data**

[Figure]

Figure 1. The geographical location of Beijing city (BJ) marked with a red star as well as surrounding regions and relevant measurement instruments implemented in this paper. The left-top panel is the topographic distribution of most of China with Beijing and surrounding areas marked, and the right-top panel is the topographic distribution of the Beijing-Tianjin-Hebei (BTH) region, with the Yanshan Mountains to the north, the Taihang Mountains to the west, and Bohai Bay to the east. The blue words are abbreviations of city names in the BTH region (a: microwave radiometer; b: 3D Doppler wind lidar; c: CIMEL sun-photometer; d: ceilometer; e: Aerodyne aerosol chemical speciation monitor (ACSM); f: multiangle absorption photometer).

**2.1 Instruments and related data**

The observation site was located at the Tower Branch of the Institute of Atmospheric Physics (IAP), Chinese Academy of Sciences (39°58′N, 116°22′E; altitude: 58 m). The IAP site is located at the intersection of the north ring-3 and north ring-4 roads in Beijing, China, among educational, commercial and residential areas, and represents a typical urban site in Beijing (hereinafter BJ site). All the sampling instruments are placed at the same location to conduct simultaneous monitoring. All the data used in this paper were recorded from July 22 to 27, 2019, and are reported in Beijing Standard Time.

Air temperature and relative and absolute humidity profiles were collected with a microwave radiometer (RPG-HATPRO-G5 0030109, Germany). The microwave radiometer (hereinafter MWR) produces profiles with a resolution ranging from 10-30 m up to 0.5 km, profiles with a resolution ranging from 40-70 m between 0.5 and 2.5 km and profiles with a resolution ranging from 100-200 m from 2 to 10 km at a temporal resolution of 1 s. A detailed description of RPG-HATPRO-type instruments can be found at http://www.radiometer-physics.de.

Vertical wind speed and horizontal wind vector profiles were retrieved with a 3D Doppler wind lidar (Windcube 100s, Leosphere, France). The wind measurement results have a spatial resolution ranging from 1-20 m up to 0.3 km and one of 25 m from 0.3 to 3 km, with a temporal resolution of 1 s. More instrument details can be found on www.leosphere.com.

A ceilometer (CL51, Vaisala, Finland) recorded atmospheric BSC profiles. The CL51 ceilometer digitally sampled the return backscattering signal from 0 to 100 μs and provided BSC profiles with a spatial resolution of 10 m from the ground to a height of 15 km. As PM mostly suspends in the ABL and is barely present in the free atmosphere, the ABLH was determined by the sharp change in the negative gradient of BSC profiles (Muenkel et al., 2007). More detailed information on ABLH calculation and screening can be found in previous studies (Tang et al., 2016; Zhu et al., 2018).

The aerosol optical depth (AOD) is observed by a CIMEL sun-photometer (CE318, France), and the AOD at 500 nm is adopted in this paper. The CE318 instrument is a

multichannel, automatic sun-and-sky-scanning radiometer and only acquires measurements during daylight hours (with the sun above the horizon). Detailed information on the AOD inversion method and the CE318 instrument have been presented in Gregory (2011).

The real-time hourly mean $PM_{2.5}$, $PM_{10}$, $O_3$, $NO_2$ and $SO_2$ ground levels were downloaded from the China National Environmental Monitoring Center (CNEMC) (available at http://106.37.208.233:20035/). All operational procedures are strictly conducted following the Specification of Environmental Air Quality Automatic Monitoring Technology (HJ/T193-2005, available at http://kjs.mep.gov.cn/hjbhbz/bzwb/dqhjbh/jcgfffbz/200601/t20060101_71675.htm). The PM chemical species, including the organics (Org), sulfate ($SO_4^{2-}$), nitrate ($NO_3^-$), ammonium ($NH_4^+$) and chloride ($Cl^-$), were measured every hour with an aerosol chemical speciation monitor (ACSM). More detailed descriptions of the ACSM have been given in Ng et al. (2011). The black carbon (BC) mass concentration was measured with a multiangle absorption photometer (MAAP5012, Thermo Electron). A more detailed description of the MAAP5012 instrument can be found in Petzold and Schonlinner (2004). As shown in Fig. 2, the ACSM-measured $PM_{2.5}$ mass concentration (=organics + sulfate + nitrate + ammonium + chloride + BC) tracked the online $PM_{2.5}$ mass concentration well, which was directly measured with a PM analyzer (from CNEMC), with a correlation coefficient ($R^2$) of 0.82. On average, the ACSM-measured $PM_{2.5}$ mass concentration accounts for 80% of the online $PM_{2.5}$ mass concentration. All chemical compositions measured by the ACSM, including organics, sulfate, nitrate ammonium and chloride, as well as BC, represent the dominant species of $PM_{2.5}$.

[Figure]

Figure 2. Scatter plot of the relationship between the directly measured PM$_{2.5}$ mass concentration (with the particulate matter analyzer of the China National Environmental Monitoring Center) and the ACSM-measured PM$_{2.5}$ mass concentration (the sum of the chemical constituent mass concentrations measured with the aerosol chemical speciation monitor (ACSM) and the black carbon (BC) mass concentration measured with the multiangle absorption photometer).

**2.2 Other datasets**

The virtual potential temperature ($\theta_V$) and pseudoequivalent potential temperature ($\theta_{se}$) are calculated by Eqs. (1) and (2), respectively:

$$\theta_v = T(1 + 0.608q)(\frac{1000}{P})^{0.286} \qquad (1)$$

$$\theta_{se} = T(\frac{1000}{P})^{0.286} exp\,(\frac{r_s L_v}{C_{pd}T}) \qquad (2)$$

where $T$ is the air temperature, $q$ is the specific humidity, $p$ is the air pressure, $r_s$ is the saturation mixing ratio, $Lv$ is the latent heat of vaporization, i.e., $2.5\times10^6$ J kg$^{-1}$, and $C_{pd}$ is the specific heat of air, i.e., 1005 J kg$^{-1}$ K$^{-1}$. All the relevant parameters can be calculated from the MWR-measured temperature and humidity profile data, and the $\theta_v$ and $\theta_{se}$ values at the different altitudes can then be further obtained. The hourly turbulence kinetic energy (TKE) is calculated as:

$$\text{TKE} = 0.5 \times (\delta_u^2 + \delta_v^2 + \delta_w^2) \qquad (3)$$

The one-hour vertical velocity standard deviation ($\delta_w^2$) and the one-hour horizontal wind standard deviation ($\delta_u^2$ and $\delta_v^2$) are calculated with Eqs. (4), (5) and (6), respectively:

$$\delta_w^2 = \frac{1}{N-1}\sum_{i=1}^{N}(w_i - \overline{w})^2 \tag{4}$$

$$\delta_u^2 = \frac{1}{N-1}\sum_{i=1}^{N}(u_i - \overline{u})^2 \tag{5}$$

$$\delta_v^2 = \frac{1}{N-1}\sum_{i=1}^{N}(v_i - \overline{v})^2 \tag{6}$$

where N is the number of records each hour, $w_i$ is the $i$th vertical wind velocity (m s$^{-1}$), $u_i(v_i)$ is the $i$th horizontal wind speed (m s$^{-1}$), $\overline{w}$ is the mean vertical wind speed (m s$^{-1}$), and $\overline{u}(\overline{v})$ is the mean horizontal wind speed (m s$^{-1}$) (Wang et al., 2019a; Banta et al., 2006). Atmospheric reanalysis data from the National Centers for Environmental Prediction (NCEP) were collected 4 times a day at 0200, 0800, 1400, and 2000 (local time) at a horizontal resolution of 2.5° × 2.5°.

**3 Results and discussion**

**3.1 Typical air pollution episodes in summer in Beijing**

[Figure]

Figure 3. (a) Temporal variations in the PM$_{2.5}$, PM$_{10}$ and O$_3$ mass concentrations as well as in the aerosol optical depth (AOD) at the BJ site from July 22-27, 2019; (b) temporal variations in the vertical atmospheric backscattering coefficient (BSC) profiles at the BJ site from July

22-27, 2019 (the yellow mark represents the light-haze pollution period, and the red mark represents the heavy-haze pollution episode).

Considering that the daily mean PM$_{2.5}$ mass concentration on both 22 July and from 26-27 July exceeded the national secondary standard (75 μg m$^{-3}$) (GB3095-2012) with maximum hourly averages up to 131 and 152 μg m$^{-3}$, respectively, two severe PM pollution processes occurred, defined as Haze I and Haze II. During these two haze periods, high atmospheric BSC levels mainly occurred below an altitude of 0.5 km, with values ranging from ~4-6 M m$^{-1}$ sr$^{-1}$. This reflects the vertical distribution of ambient particles from the aspect of aerosol scattering to a certain degree, namely, only do the suspended particles concentrated in the lower layer of the atmosphere. According to the National Ambient Air Quality Standards (GB3095-2012), an ozone pollution day is any day when the hourly mean O$_3$ concentration is higher than 160 μg m$^{-3}$; thus, during the observation periods, each day was a severe ozone pollution day. As reported by the Ministry of Ecology and Environment, in 2018, the number of motor vehicles reached 327 million, up by 5.5% year-on-year (available at http://www.mee.gov.cn/xxgk2018/xxgk/xxgk15/201909/t20190904_732374.html). Although stringent pollution control measures have been implemented regarding factories, motor vehicles still discharge large amounts of primary pollutants into the atmosphere, including NOx, HC, VOCs, and CO. Under high solar radiation and temperature levels in summer, photochemical processes are prominent, contributing to a high O$_3$ concentration along with many highly reactive radicals, which further enhance the oxidizing capacity of the atmosphere (Frischer et al., 1999; Sharma et al., 2013). Haze pollution under a high atmospheric oxidation capacity had likely occurred on 22 July and from 26-27 July. Generally, due to the stringent pollutant emission control measures, the emission of primary aerosols is low, with a very low PM$_{2.5}$ level in summer in Beijing. The sudden elevated ambient particle concentration (the Haze I and Haze II periods) resulted in the worst PM pollution in Beijing that summer and has been widely concerned by the public. Thus, the formation mechanism of the Haze I and Haze II periods during which the PM$_{2.5}$ and O$_3$ concentrations were simultaneously/alternately high should be systematically examined. The key point is to

determine the oxidation capacity of the regional atmosphere and to clarify the formation mechanism of secondary aerosols. In addition, the occurrence and evolution patterns of these two haze processes were different, which could refer to the diverse accumulation mechanisms, regional transfer contributions, ABL structures, and removal processes. Therefore, by clarifying the various pollution processes, it should be possible to determine the leading factors of these haze phenomena in Beijing in summer. In summary, we will examine the haze pollution causes under a high atmospheric oxidization capacity in terms of the physical processes, such as pollutant sources and sinks and ABL structure influence, and chemical processes, namely, aerosol transformation processes.

**3.2 The formation mechanism of haze pollution in summer in Beijing**

[Figure]

Figure 4. Temporal variation in the vertical profiles of the (a) horizontal wind vector (the white arrows denote wind vectors), (b) temperature (T), (c) absolute humidity (AH), and (d) relative humidity (RH) at the BJ site from July 22-27, 2019 (the yellow mark represents the light-haze pollution period, and the red mark represents the heavy-haze pollution episode).

[Figure]

Figure 5. Temporal variation in the vertical profiles of the (a) virtual potential temperature gradient (VPTG: $\partial\theta_v/\partial z$), (b) pseudoequivalent potential temperature gradient (EPTG: $\partial\theta_{se}/\partial z$) and (c) turbulent kinetic energy (TKE), along with the corresponding (d) atmospheric boundary layer height (ABLH) at the BJ site from July 22-27, 2019 (the yellow mark represents the light-haze pollution period, and the red mark represents the heavy-haze pollution episode).

[Figure]

Figure 6. Temporal variation in the (a) PM$_{2.5}$, O$_3$, NO$_2$ and SO$_2$ mass concentrations, (b) PM$_{2.5}$

chemical composition, including organics (Org), sulfate ($SO_4^{2-}$), nitrate ($NO_3^-$), ammonium salt ($NH_4^+$), chlorine salt ($Cl^-$) and black carbon (BC) at the BJ site from July 22-27, 2019. (c) Temporal variation in the relative contributions of the chemical components to the $PM_{2.5}$ mass concentration at the BJ site from July 22-27, 2019. (d) Temporal variation in the mass concentrations of the dominant $PM_{2.5}$ chemical components, sulfur oxidation ratio (SOR) and nitrogen oxidation ratio (NOR) at the BJ site from July 22-27, 2019. (e) Temporal variation in the relative humidity (RH) and temperature (T) at the BJ site from July 22-27, 2019 (the yellow mark represents the light-haze pollution period, and the red mark represents the heavy-haze pollution episode).

**3.2.1 The occurrence stage**

Fig. 3a reveals that the PM level in Beijing gradually increased from 8:00 to 22:00 on 26 July (the Haze II episode) and from 4:00 to 22:00 on 22 July (the Haze I episode), with the $PM_{2.5}$ mass concentration eventually reaching 75 and 131 µg m$^{-3}$, respectively. These two stages are regarded as the Haze I and Haze II occurrence stages.

**a. *The contribution of pollution transport**

Owing to the notable control measures in summer in Beijing, the sudden elevated PM levels very likely originated from an outside region. Clearly, since the early hours on 22 and 26 July, Beijing was located behind the northwest-southeast trough of the 850-hPa potential height field, which bordered the Sichuan Basin to the west (Fig. 7a-d; Fig. 7i-l). Therefore, Beijing was always controlled by strong southerly winds at high altitudes. With the Taihang Mountains to the east and the Yanshan Mountains to the north (Fig. 2), Beijing is a semi-enclosed area; thus, the south wind belt passing through the North China Plain to Beijing will be strengthened (Su et al., 2004). The southerly wind speeds ranged from ~8-10 m s$^{-1}$ (the Haze II period) and from ~5-7 m s$^{-1}$ (the Haze I period) at altitudes >0.5 km. As strong southerly winds persisted from the south, a moisture transport channel, with southerly winds carrying water vapor to Beijing, was formed and increasingly intensified (Fig. 7a-d; Fig. 7i-l). In response, the humidity in Beijing conspicuously increased in the morning of 26 July with the AH (RH) ranging from ~15-17 g m$^{-3}$ (~75%), while the AH (RH) decreased to ~13 g m$^{-3}$ (~70%) from 10:00 on (Fig. 4c-d). The air temperature during the daytime was extremely high,

ranging from ~30-35 °C (Fig. 4b), and these high-temperature weather conditions reduced the humidity by evaporation to a certain degree. Considering that the air temperature was always very high (~30 °C) since the early morning on 22 July, the AH (RH) was ~13 g m$^{-3}$ (~65%) during the occurrence stage.

With the more densely populated industrial regions located in the south of Beijing, the strong winds blowing from the south were also highly likely to transport large amounts of anthropogenic aerosols to Beijing (Chang et al., 2018; Liu et al., 2013b). To examine the potential PM transportation, we generated PM$_{2.5}$ mass concentration distribution maps for most parts of China (Fig. 8) and combined them with corresponding background circulation fields to elucidate the pollution transportation phenomenon. The regional distribution of the PM$_{2.5}$ mass concentration was obtained by interpolating PM$_{2.5}$ data from more than 1000 stations of the China National Environmental Monitoring Centre into a grid (0.5°×0.5°). Notably, at 2:00 on 26 and 22 July, high PM$_{2.5}$ mass concentrations (~70 μg m$^{-3}$ during the Haze I episode and ~50 μg m$^{-3}$ during the Haze II episode) mainly occurred in the south/southwest area of Beijing, which were substantially higher than that in Beijing city (~10 μg m$^{-3}$) (Fig. 8a-b; Fig. 8i-l). The heavily polluted southern area of Beijing mainly included Baoding, Langfang and Shijiazhuang, which are generally ~60-300 km away from Beijing (Fig. 2). The southerly air mass above ~0.5 km moved faster than ~20-30 km h$^{-1}$ (estimated from the measured wind speed) on 26 and 22 July, which was fast enough to transport pollutants to Beijing in several hours. As expected, the area with a high PM$_{2.5}$ mass concentration gradually spread northward corresponding to the southerly winds, and consequently, the highest PM$_{2.5}$ level occurred in Beijing at 20:00 on both 26 July (reaching ~65 μg m$^{-3}$) and 22 July (reaching ~80 μg m$^{-3}$). This was consistent with the PM$_{2.5}$ increase trends at this time, as shown in Fig. 3a. The average increase rate of the PM$_{2.5}$ concentration (~5.8 μg m$^{-3}$ h$^{-1}$) on 22 July was higher than that on the 26 July (~3.73 μg m$^{-3}$ h$^{-1}$), possibly related to the large difference in the PM$_{2.5}$ concentration between Beijing city and the southern area of Beijing. These results are consistent with the findings reported by Zhong et al. (2019). Thus, multiple results implied that PM transportation by southerly winds was primarily responsible for the PM increase at the

occurrence stage.

**b.  *The effect of the atmospheric boundary layer structure**

As shown in Fig. 5a-b, in the mornings on 26 and 22 July, the positive values of the virtual potential temperature gradient ($\partial\theta_v/\partial z$) and pseudoequivalent potential temperature gradient ($\partial\theta_{se}/\partial z$) at altitudes ranging from 0-2 km (the Haze II period) and from 0-1 km altitude (the Haze I period) indicated that a stable atmosphere layer was present. Generally, with no solar radiation reaching the ground and more upward longwave radiation emitted from the ground at night, the surface cools faster than the upper atmosphere, thus promoting a stable atmosphere. In response, the turbulent kinetic energy (TKE) was extremely low (0-1 $m^2$ $s^{-2}$) along with a low ABLH of ~0.5 km (Fig. 5c-d). This means that on both 26 and 22 July, south winds persisted as the ABL structure was not conducive to vertical substance diffusion. The stable ABL structure suppressing vertical pollution diffusion also contributed to the occurrence of PM pollution to a certain degree. Both $\partial\theta_v/\partial z$ and $\partial\theta_{se}/\partial z$ at an altitude ranging from 0-1.5 km became negative from 14:00-16:00 on 26 July, indicating an unstable atmosphere layer. Generally, the high daytime solar radiation reaching the surface may rebuild the vertical temperature structure and disrupt the stable ABL, especially in summer (Andrews, 2000). Thus, turbulence was quickly generated by the thermodynamic activity with the TKE increasing to ~2-3 $s^2$ $m^{-2}$ and continuing to develop upwards, causing the ABLH to gradually increase to ~1.2 km. This ABL process explained the slight fluctuations in the PM increase at this time in which the $PM_{10}$ mass concentration sharply decreased from 100 to 73 $\mu g$ $m^{-3}$. In contrast to the ABL condition on 26 July (the Haze II stage), $\partial\theta_{se}/\partial z$ was negative, but $\partial\theta_v/\partial z$ was positive below ~1.5 km in the afternoon on 22 July (the Haze I stage). Combined with a low TKE (~0-0.5 $m^2$ $s^{-2}$) similar to that in the morning, the atmospheric stratification below ~1.5 km remained absolutely stable. Maybe due to the low solar radiation gradually heating the ground in the afternoon under cloudy weather conditions, the original stable ABL structure previously formed in the nighttime could not be disrupted. All the above results imply that the ABL structure also plays a role in the PM increase at the occurrence stage.

***c. Secondary aerosol formation driven by a high atmospheric oxidation capacity***

When the PM$_{2.5}$ concentration increased due to the strong southerly winds in Beijing during the Haze II (Haze I) occurrence stage, O$_3$ increased sharply, rapidly increasing from 67 (26) µg m$^{-3}$ and peaking at 250 (131) µg m$^{-3}$. As mentioned in section 3.1, a high O$_3$ concentration indicates a high atmospheric photochemical reactivity (Li et al., 2012; Seinfeld, 1986); thus, the atmosphere had a high oxidizing capacity with large amounts of free radicals (OH, etc.) and ozone, which promoted secondary aerosol formation (Pathak et al., 2009; Shi et al., 2015; Wang et al., 2016). Fig. 6b shows that along with the increase in PM$_{2.5}$ concentration during the occurrence stage, the organics, sulfate, and nitrate concentrations in PM$_{2.5}$ also gradually increased. The average organics, sulfate, and nitrate concentrations during the Haze II (Haze I) occurrence stage were 15.6 (23.0) µg m$^{-3}$, 10.0 (8.0) µg m$^{-3}$ and 4.3 (24.7) µg m$^{-3}$, respectively, and accounted for 40.7 (32.1)%, 25.3 (11.2)%, and 12.2 (31.5)%, respectively, of the PM$_{2.5}$ concentration. The total sulfate, organics, and nitrate (SON) concentration accounted for more than 75% of the PM$_{2.5}$ concentration during both the Haze II and Haze I occurrence stages (Fig. 6c), implying that the SON increase was the leading cause of the PM$_{2.5}$ concentration increase. Secondary organic aerosols can be formed by the photochemical oxidation reactions of the VOCs emitted by vehicles (Hennigan et al., 2011). Thus, the high concentration and relative contribution of organics are mainly attributed to the notably active photochemical reactions in summer and high VOCs emissions by vehicles in Beijing city. Due to the lack of VOCs data, the detailed formation mechanism of secondary organics will be studied in the future. To examine the possible formation mechanism of secondary inorganic aerosols, the sulfur oxidation ratio (SOR) and nitrogen oxidation ratio (NOR), defined as $SOR = [SO_4^{2-}]/([SO_4^{2-}] + [SO_2])$ and $NOR = [NO_3^-]/([NO_3^-] + [NO_2])$, respectively, where $[\ ]$ indicates the molar concentration, were adopted in this paper. Higher SOR and NOR values suggest a higher oxidation efficiency of sulfur and nitrogen, which means that more secondary inorganic aerosols occur in the atmosphere (Liu et al., 2019c; Han et al., 2019; Yao et al., 2002; Kong et al., 2018; Sun et al., 2006).

Both homogeneous gas-phase and heterogeneous reactions can promote the

formation of sulfate from $SO_2$ during haze episodes (Khoder, 2002; Harris et al., 2013), thereby increasing the SOR. Notably, the SOR values during the whole observation period (from 22 to 27 July) were relatively high, averaging 0.62, along with relatively low $SO_2$ levels, averaging 2.2 μg m$^{-3}$ (Fig. 6a; d). The observed high SOR values could be attributed to the relatively high RH (an average of ~66.6%) (Fig. 6e) and the ubiquitous photochemical reactions in summer in Beijing (Han et al., 2019). Nevertheless, compared to the very low PM level on clean days (on 25 July) (Fig. 6d), the temporal variation in the sulfate concentration on 26 July (the Haze II period) and 22 July (the Haze I period) exhibited a distinct increasing trend during the occurrence stage, gradually increasing from 3.7 to 14.4 μg m$^{-3}$ and from 4.2 to 11.5 μg m$^{-3}$, respectively. Moreover, the SOR values also averaged ~0.76 at higher levels during both the Haze II and Haze I occurrence stages compared to clean days, which attained an average of ~0.55 (Fig. 6c). The results indicated enhanced secondary sulfate aerosol formation during the occurrence stage. However, the PM level and sulfate concentration on clean days were very low, but the $O_3$ concentration was relatively high (Fig. 6a), reaching up to 214 μg m$^{-3}$, which implied highly active photochemical reactions. Thus, although the notable photochemical reactions occurring during the daytime on 26 and 22 July facilitated homogeneous gas-phase $SO_2$ oxidation to a certain extent, it was not the dominant reason for the sulfate increase during the occurrence stage. Notably, the PM level and total chemical component mass concentration slowly increased on 24 July with no pollution transportation by south winds (Fig. 3a-b; Fig. 7e-h; Fig. 8e-h;), while the average sulfate concentration was 2.8 μg m$^{-3}$ and only accounted for 10.7% of the $PM_{2.5}$ concentration, far lower than that during the Haze II period and similar to that on clean days. The average RH was 61.4% and 75.3% during the Haze II and Haze I occurrence stages, respectively, which was also higher than that on clean days (54.5%). According to the results mentioned above, the strong winds blowing from the south and southwest of Beijing transport much moisture and particles, and we infer that the increase in sulfate aerosols during the Haze II and Haze I periods can be mainly attributed to regional transport. Hence, the moisture and particles transported to Beijing further facilitated the heterogeneous reactions of $SO_2$ on the moist aerosol surface. This

highlights the importance and urgency of enhancing joint regional pollution emission control.

Nitrate is predominantly formed via both the homogeneous gas-phase photochemical reaction of $NO_2$ with OH radicals in the daytime when the photochemical activity is high (Wang et al., 2006; Wen et al., 2018; Seinfeld and Pandis, 2006) and the heterogeneous hydrolysis reaction of $NO_3$ and $N_2O_5$ in the atmosphere in the nighttime (Richards, 1983; Russell et al., 1986; Wang et al., 2009; Wang et al., 2017a; Pathak et al., 2011). In addition, there exists an equilibrium between particulate nitrate and gaseous $HNO_3$ and $NH_3$ in the atmosphere because ammonium nitrate is semi-volatile (Seinfeld, 1986). A high temperature could promote ammonium nitrate decomposition; thus, the regional transport of ammonium nitrate in summer was not considered (Li et al., 2019). Fig. 6b and d reveal that the nitrate concentration (NOR) during the occurrence Haze II stage slightly increased from 3.2 μg m$^{-3}$ (0.09) at 8:00 to 5.2 μg m$^{-3}$ (0.23) at 22:00. The nitrate concentration (NOR) during the Haze I occurrence stage sharply increased from 2.7 μg m$^{-3}$ (0.02) at 8:00 to 38.1 μg m$^{-3}$ (0.36) at 16:00. The nitrate concentration and relative contribution to PM during the Haze I period were markedly higher than those during the Haze II period (Fig. 6c). This inconsistency could be attributed to the higher temperature (averaging ~34 °C) during the Haze II period than that during the Haze I period (averaging ~27 °C) (Fig. 6e). These results indicated that strong photochemical reactions facilitated nitrate formation, thereby increasing the PM$_{2.5}$ level, while nitrate decomposed into gaseous $HNO_3$ and $NH_3$ once the temperature was high enough. After 15:00, the nitrate concentration increased in the presence of large amounts of radicals, and the temperature drop inhibited the reverse reaction. In the nighttime, the increase in nitrate aerosols was predominantly attributed to the heterogeneous hydrolysis reactions of $NO_3$ and $N_2O_5$ in the atmosphere; more details are provided in the next section.

**3.2.2 The outbreak stage**

The PM$_{2.5}$ mass concentration suddenly increased from 75 μg m$^{-3}$ at 22:00 on 26 July to 146 μg m$^{-3}$ at 4:00 on 27 July and remained high at ~150 μg m$^{-3}$ until 10:00, which was identified as the outbreak stage of haze pollution (Fig. 3a). Compared to

the atmospheric BSC ranging from ~2.5-3 M m$^{-1}$ sr$^{-1}$ on 26 July, the ambient particle concentration below the ~0.5-km altitude sharply increased the atmospheric scattering coefficient, exceeding 6 M m$^{-1}$ sr$^{-1}$ (Fig. 3b).

**a. The almost negligible contribution of southerly transport**

There were still strong southerly winds controlling Beijing at high altitudes (>0.5 km), accompanied by a more notable vapor transportation channel below (Fig. 7m-n). However, the PM levels in the south/southeast area of Beijing, ranging from 0 to ~60 μg m$^{-3}$, were significantly lower than those (>80 μg m$^{-3}$) in Beijing, even below air quality standards (Fig. 8n-m). It was unlikely that the explosive PM growth and the persistent high PM level in Beijing were caused by pollution transportation.

**b. Extremely stable ABL structures are a prerequisite for pollution outbreaks**

Without pollution transportation, more attention was focused on the interior of the local ABL, and Fig. 5 shows the temporal variation in the ABL structure. Both the $\partial\theta_v/\partial z$ and $\partial\theta_{se}/\partial z$ values became positive (~1.5 °C/100 m and ~2.5 °C/100 m, respectively) below the ~0.3-km altitude, as depicted in Fig. 5a-b. This implied that a very stable lower layer defined as the nocturnal stable boundary layer (NSBL) had formed with an ABLH of ~0.3 km. Due to the notable radiation effect of the already high aerosol loading during the daytime, the surface solar radiation was greatly blocked and reduced, which promoted stable stratification at midnight (Zhao et al., 2019; Zhong et al., 2017). In such a thermally stable state, the buoyancy transport heat flux in the atmosphere continuously consumes turbulent energy, suppressing the development of turbulence. Therefore, the corresponding TKE had sharply decreased compared to that from 14:00-16:00 on 26 July, lower than ~0.5 m$^2$ s$^{-2}$ and even approaching ~0 m$^2$ s$^{-2}$ (Fig. 5c-d). However, the $\partial\theta_v/\partial z$ and $\partial\theta_{se}/\partial z$ values were positive and negative, respectively, from ~0.3 to ~1.5 km, which implies that this atmospheric layer was conditionally instable. Considering the very low TKE like that below ~0.3 km, this layer, referred to as the residual layer, was also absolutely stable. Thus, the ambient particles were restrained from vertically spreading and were concentrated below the NSBL, thereby increasing the ground PM level. The same work would happen to the ambient water vapor transported by the southerly winds, which

explained the extremely high humidity during this period. As shown in Fig. 4c-d, the atmospheric humidity during the outbreak stage was distinctly higher than that on 26 July with the AH (RH) reaching ~20 g m$^{-3}$ (~90%). In contrast to the role of the moisture transport channel, the unique NSBL structure has a more notable impact on the increase in air humidity.

In contrast, during the Haze I period on 22 July, no PM pollution outbreak stage occurred, as the PM$_{2.5}$ mass concentration had sharply decreased from 131 to 53 μg m$^{-3}$ in one hour since 21:00. The ambient particles did not accumulate and maintained a high level, similar to that during the Haze II period, because the ABL structure did not exhibit similar characteristics. The already high PM$_{2.5}$ level (~130 μg m$^{-3}$) in the daytime accelerated surface cooling, causing the NSBL to more readily form at a very low height of ~0.2 km. This situation was similar to that during the Haze II episode. Nevertheless, the TKE above the NSBL was very high, reaching ~2-3 m$^2$ s$^{-2}$, in notable contrast to that during the Haze II episode, where the TKE was extremely low (~0 m$^2$ s$^{-2}$) across the whole 0-1.5 km layer. The vertical temperature structures above the NSBL indicated that the atmosphere had attained conditional instability, while in terms of the TKE distribution, the atmospheric stratification above the NSBL during the Haze I period was unstable, in contrast to the stable stratification during the Haze II period. Because it rained at night with a high AH (~15-20 g m$^{-3}$) and RH (>90%) extending from the surface up to an altitude of ~3 km, the convection activity was quite strong accompanied by a wet deposition process. Due to the unstable ABL structure and the accompanying wet deposition, the ambient particle concentration did not sharply increase, but particles were instead removed from the atmosphere.

Noted that the PM level on 24 July also tended to increase, but it suddenly decreased, similar to that during the Haze I stage. There was no transportation effect contributing to the increase in PM level on 24 July under westerly circulation field control (Fig. 7e-h). Similar to the Haze I and Haze II occurrence stages, a stable atmosphere near the surface had formed with positive $\partial\theta_v/\partial z$ and $\partial\theta_{se}/\partial z$ values. Under this stable stratification, the PM from local emissions started increasing on 24 July. Because of the high daytime solar radiation quickly heating the surface, the anomalous

vertical temperature structures formed by longwave radiation cooling during the nighttime were disrupted and transformed into unstable stratifications with negative $\partial\theta_{se}/\partial z$ ($\partial\theta_v/\partial z$) profiles. As observed during the Haze II episode, the ABL structure characterized by an increased TKE ($\sim$2-3 m$^2$ s$^{-2}$) and elevated ABLH ($\sim$1.5 km) resulted in rapid pollution dissipation. However, the difference between the Haze II process and the pollution process on 24 July was that the unstable atmospheric stratification with a high TKE on 24 July continued to develop until the end of the day, while for the Haze II process, this condition lasted only two or three hours at noon. Additionally, an NSBL was established at midnight during the Haze II process at an ABLH of $\sim$0.3 km, thus worsening the near-stratum vertical diffusion conditions. Therefore, the subsequent stable atmospheric stratification on 26 July was a prerequisite for the pollution outbreak during the Haze II process. Particles would not accumulate and cause pollution outbreak without a stable ABL structure but were easily removed by the self-cleaning capacity of the atmosphere.

**c. Intense secondary aerosol formation driven by the atmospheric oxidation capacity causing the pollution outbreak**

Heterogeneous aqueous reactions refer to the secondary formation of sulfates and nitrates largely related to the ambient humidity (Wang et al., 2012). The accumulation of water vapor in the NSBL facilitated secondary aerosol formation and further promoted the outbreak of PM pollution. To investigate the explosive growth mechanisms, we divided the PM pollution outbreak stage during the Haze II process into two stages: stage I, from 22:00 on 26 July to 4:00 on 27 July; stage II, from 5:00 to 10:00 on 27 July. During stage I, along with the explosive growth in PM$_{2.5}$, the nitrate concentration rapidly increased from 11.6 to 57.8 $\mu$g m$^{-3}$, while sulfate and organics slightly increased from 13.7 to 19.8 $\mu$g m$^{-3}$ and from 21.8 to 24.9 $\mu$g m$^{-3}$, respectively (Fig. 6d). During stage II, the nitrate concentration remained at its highest level of $\sim$57 $\mu$g m$^{-3}$, and the sulfate level remained at $\sim$19 $\mu$g m$^{-3}$, with the organics slowly decreasing (Fig. 6d). The explosive growth trend of nitrate is the most consistent with that of PM$_{2.5}$. In addition, the average organics, sulfate, and nitrate concentrations during the whole outbreak stage were 20.6, 15.9 and 43.0 $\mu$g m$^{-3}$, respectively, and

accounted for 22.0%, 17.8%, 34.9%, respectively, of the $PM_{2.5}$ concentration. Compared to the occurrence stage, the relative contributions of organics and sulfate to $PM_{2.5}$ decreased significantly, while the contribution of nitrate notably increased. These results indicated that the explosive $PM_{2.5}$ concentration growth was driven by the sharp increase in nitrate concentration. With strong photochemical reactions during the daytime, the $O_3$ mass concentration was very high before the outbreak stage, up to 214 $\mu g\ m^{-3}$. $NO_2$ was produced by $O_3$ reacting with a large amount of NO, which was discharged by vehicles during evening hours. $NO_2$ reacted with $O_3$ aloft to form $NO_3$, which rapidly reacted with $NO_2$ to form $N_2O_5$ at night. During stage I, NOR rapidly increased from 0.26 to 0.60, which implied that the $NO_2$ oxidization rate sharply increased within a few hours. Considering that $NO_2$ remained relatively low at ~25 $\mu g$ $m^{-3}$ and $O_3$ rapidly decreased from 214 to 46 $\mu g\ m^{-3}$ during stage I (Fig. 6a), the consumption process of $NO_2$ was more significant than its generation process. The $NO_2$ produced through $O_3$ consumption was constantly oxidized by $O_3$ to generate a large amount of $N_2O_5$, resulting in a sharp decline in the $O_3$ concentration. Once $N_2O_5$ was produced, it would be adsorbed onto moist particle surfaces and react with water droplets to form nitrate, resulting in a sudden nitrate increase, from 11.6 to 57.8 $\mu g\ m^{-3}$. During stage II, $O_3$ slowly decreased to 34 $\mu g\ m^{-3}$ at 6:00 on 27 July, and $NO_2$ remained relatively high (~44-51 $\mu g\ m^{-3}$), which meant that the $NO_2$ generation process dominated. Thus, the oxidization of $NO_2$ did not further increase as the NOR remained at ~0.45 during stage II. Hence, nitrate, formed along the pathway whereby $N_2O_5$ was adsorbed onto surfaces and reacted with water droplets, did not further increase, maintaining its highest mass concentration of ~57 $\mu g\ m^{-3}$. The processes mentioned above were unimportant during the daytime because $N_2O_5$ was in equilibrium with $NO_3$; that is, $NO_3$ was photolyzed and rapidly destroyed by NO, which in turn occurred whenever $NO_x$ and sunlight were present. During both stages I and II, the SOR always remained relatively high at ~0.95, accompanied by a high RH of ~90%. A high SOR and RH signified that heterogeneous reactions dominated the formation of particulate sulfate during the outbreak stage. The increased sulfate amount, which was lower than that of nitrate, may be related to the low $SO_2$ emissions and massive NO emissions from

the large number of vehicles. This highlights the importance and urgency of enhancing NOx (vehicle) emission control.

Contrary to expectations, after the wet deposition process during the Haze I period, the $PM_{2.5}$ and $NO_2$ concentrations and the total chemical composition abruptly increased at 0:00 on 23 July, accompanied by a sharp increase in nitrate and NOR (from 9.3 to 41.5 μg m$^{-3}$ and from 0.26 to 0.49, respectively). These results may be related to the high RH (higher than 93%), which facilitated the heterogeneous hydrolysis reaction of $NO_3$ and $N_2O_5$, formed from gas pollutants NOx and $O_3$ not completely removed in the wet deposition process.

**3.2.3 The diffusion stage**

After 10:00 on 27 July, the $PM_{2.5}$ mass concentration sharply decreased to 50 μg m$^{-3}$ over three hours, during which the atmospheric BSC decreased to $<1\times10^3$ M m$^{-1}$ sr$^{-1}$ across the whole ABL (Fig. 3 and Fig. 8o-p). This represented the pollution diffusion stage. As no wet deposition process occurred, the Haze II diffusion stage was different from that of Haze I. Generally, the arrival of strong and clean air masses from the south is the main factor dissipating air pollution in Beijing (Zhong et al., 2017; Zhong et al., 2018; Zhao et al., 2019). Calm/light winds in the lower layer dominated during the outbreak stage, while sudden increased southeast winds persisted in the 0-2 km layer after 8:00 on 27 July, with a wind speed of ~6-9 m s$^{-1}$ (Fig. 7n-q and Fig. 4a). The southeast winds originated from the Bohai Sea and the Yellow Sea. Moreover, during this diffusion stage, the air quality of the southeast of Beijing was basically clean or much better than that in Beijing (Figure 8(n)-(p)). Therefore, strong southeast winds would not bring pollutants aggravating the pollution in Beijing instead played a role in the horizontal diffusion of the accumulated PM at the surface. On the other hand, accompanied by the horizontal diffusion, the strong solar radiation at noon reached the surface and changed the vertical temperature structure. The ABL was extremely unstable in terms of both $\partial\theta_v/\partial z$ and $\partial\theta_{se}/\partial z$, which were negative below ~1.0 km with values of -0.5 °C/100 m and -2.5 °C/100 m, respectively (Fig. 5a-b). Along with this instability, the development of turbulence in the ABL was very strong and quick, with the TKE suddenly increasing to ~3-5 m$^2$ s$^{-2}$ (Fig. 5c). Accompanied by pronounced

turbulence development, the ABL continuously developed upward with the ABLH up to ~2.5 km over a short time (Fig. 5d). The ABL structure quickly became extremely suitable for vertical pollutant diffusion; thus, the PM level sharply decreased during this time.

In contrast to PM$_{2.5}$, the O$_3$ concentration rapidly increased with increasing radiation, along with the high NO$_2$ and NO concentrations attributed to morning traffic emissions. Along with the decline in PM$_{2.5}$, organics and sulfate slowly decreased to below ~3 μg m$^{-3}$, and nitrate decreased to below 1.0 μg m$^{-3}$. The average organics, sulfate, and nitrate concentrations were as low as 6.8, 6.2 and 1.9 μg m$^{-3}$, respectively, and accounted for 33.0%, 32.3%, and 6.0%, respectively, of the PM$_{2.5}$ concentration. As the significant turbulence activity caused vertical transportation of vapor, heat, and particles, the RH decreased to ~60%, accompanied by a decline in SOR (~0.75). This emphasized the notable correlation between the humidity and the heterogeneous formation mechanism of sulfate. In addition, the NOR rapidly decreased from 0.22 to 0.01, coinciding with the change in nitrate. At this stage, the temperature always remained high at ~35 °C. Thus, similar to the occurrence stage, ammonium nitrate evaporated at high temperatures, contributing to a decline in nitrate. In summary, during the diffusion stage, the unstable ABL structure was not only conducive to pollution diffusion but also affected T and RH to inhibit secondary aerosol formation and further reduced secondary aerosols.

Regardless of the wet deposition process during the Haze I period or the horizontal and vertical diffusion during the Haze II period, air pollution eventually dissipated as long as the atmosphere was in a specific state. In other words, this implies that the self-cleaning capacity of the atmosphere was responsible for air pollution dispersion. When the atmosphere attains a specific state, its self-cleaning capacity removes pollution. To examine this phenomenon, the key factors characterizing the self-cleaning capacity of the atmosphere should be determined first. As analyzed above, once the TKE increased to >1.5-2 m$^2$ s$^{-2}$, the ABLH increased and exceeded ~1 km, and the $\partial\theta_v/\partial z$ and $\partial\theta_{se}/\partial z$ values became negative, as well as no calm/light winds persisted. The atmosphere was instable with notable turbulence activities and advection transport, and air pollution was

immediately dissipated. Owing to the limited observation time, the results regarding the characteristics of the self-cleaning capacity of the atmosphere may not be universal, and a more comprehensive investigation on the self-cleaning capacity of the atmosphere will be conducted in the future.

[Figure]

Solid line: 850-hPa geopotential height field (units: m); white arrow: wind vector (units: m s⁻¹); shaded color: 850-hPa specific humidity field (units: g kg⁻¹)

Figure 7. Composites of the 850-hPa horizontal wind vector field (units: m s⁻¹; white arrows), 850-hPa geopotential height field (units: m; solid lines) and 850-hPa specific humidity field (units: g kg⁻¹; shaded colors) at 0200, 0800, 1400, and 2000 (local time) on 22 and 24 July and from 26-27 July, labeled as (a) - (p). The star shows the location of the BJ site.

[Figure]

Figure 8. The PM$_{2.5}$ mass concentration distribution (units: μg m$^{-3}$; shaded colors) over most of China at 0200, 0800, 1400, and 2000 (local time) on 22 and 24 July and from 26–27 July, labeled as (a)–(p).

**4 Conclusion**

[Figure]

Figure 9. Schematic diagram for the formation mechanism of haze pollution under a high

atmospheric oxidization capacity in summer in Beijing (blue dashed line: atmospheric boundary layer; red solid lines: potential temperature gradient profiles; brown solid line: temporal change curve of the ozone concentration; cyan solid line: temporal change curve of the PM$_{2.5}$ mass concentration; gray arrow sectors: temporal change in the wind vector profiles; TKE: turbulence kinetic energy; solid dots: particulate matter in the atmosphere; droplets: water vapor).

The extremely serious haze pollution episode characterized by alternating/synchronous heavy PM loadings and high ozone concentrations occurred this summer in Beijing. Combined with a series of observations, the formation mechanism of haze pollution under a high atmospheric oxidization capacity has been systematically analyzed in terms of the atmospheric physical and chemical processes and schematically depicted in Fig. 9. The occurrence of haze pollution in summer in Beijing was mainly attributed to southerly transport and influenced by the ABL structure to a certain degree (physical process), which was further promoted by intense secondary aerosol formation under a high atmospheric oxidation capacity (chemical process). On the one hand, the physical process, where large amounts of moisture and particles were transported to Beijing by strong southerly winds, caused haze pollution initiation in Beijing, consistent with previous studies, e.g., Huang et al. (2017) and Zhong et al. (2019). Moreover, we found that haze pollution occurred when the ABL structure was extremely stable with a low TKE and a positive potential temperature gradient ($\partial\theta/\partial z$), which increased the PM level in Beijing. The stable ABL was disrupted and transformed into an unstable structure (negative $\partial\theta/\partial z$) with high solar radiation in the afternoon (Andrews, 2000), responsible for the fluctuations in the PM increase process. On the other hand, the concentration of secondary aerosols such as sulfate, nitrate, and organics quickly increased. The very high O$_3$ concentration in the daytime indicates an active atmospheric photochemical reactivity (Li et al., 2012; Seinfeld, 1986) and a high atmospheric oxidizing capacity with large amounts of free radicals (OH, etc.) and ozone, which promotes secondary aerosol formation (Pathak et al., 2009; Shi et al., 2015; Wang et al., 2016). However, we found that the distinct increase in sulfate concentration was mainly linked to southerly transport, which carried

heavy sulfate aerosol loadings to Beijing. The physical process, where the extremely stable ABL inhibited PM and moisture diffusion, thus increasing the ambient humidity and ground-level $PM_{2.5}$, was a prerequisite for haze pollution outbreak. Under a stable ABL, secondary aerosol formation dominated by nitrate was quite intense, driving the pollution outbreak. The PM levels in the south/southeast area of Beijing were significantly lower than those in Beijing, even below air quality standards. The contribution of pollution transport was negligible. Owing to the already high $PM_{2.5}$ level during the daytime, the strong aerosol radiation effect cooled the surface and heated the above layer (Dickerson et al., 1997; Stone et al., 2008; Wilcox et al., 2016), which facilitated NSBL formation. The $\partial\theta/\partial z$ value in the NSBL was thus found to be positive, thus increasing the atmospheric stability, decreasing the ABLH and decreasing the TKE. The ambient particles and moisture would be restrained from vertically spreading and became concentrated below the NSBL (Stone et al., 2008), resulting in elevated PM and humidity levels at the surface. In addition, there was a large increase in NOR and an explosive growth in the nitrate concentration during the outbreak stage. Due to the high $O_3$ level produced by the intense photochemical reactions during the daytime and the NOx discharged by vehicles during evening peak hours, vast amounts of $N_2O_5$ and $NO_3$ were formed through oxidization reactions (Chang et al., 1967; Wilson Jr et al., 1972). Under a very high humidity, the heterogeneous hydrolysis reactions of $N_2O_5$ and $NO_3$ at the moist particle surface were very notable, resulting in the formation of large amounts of nitrate aerosols (Richards, 1983; Russell et al., 1986; Wang et al., 2009; Wang et al., 2017a; Pathak et al., 2011). Considering that pollutant transport from outside considerably affected haze formation in Beijing, especially during the occurrence stage, continuous regional joint control of air pollution should be enhanced. In addition, as reported in previous studies (Li et al., 2012; Pathak et al., 2009; Seinfeld, 1986; Shi et al., 2015; Wang et al., 2016; Zhong et al., 2018) and confirmed in this study, the atmospheric oxidization capacity, enhanced by photochemical reactions, largely facilitated secondary aerosol formation, which further aggravated pollution. In this study, secondary organic aerosols and secondary nitrate aerosols significantly increased and were the most important constituents of particles during the haze episodes.

Photolysis of NOx triggers photochemical reactions, in which the reactions with VOCs are important (Hennigan et al., 2011; Seinfeld and Pandis, 2006; Wang et al., 2006; Wen et al., 2018). Additionally, NOx and VOCs are precursors of nitrate and organics, respectively. Thus, controls should be strengthened for supervising heavy diesel vehicles and collaboratively controlling NOx and VOC emissions. As the PM level gradually increased, a wet deposition process and an extremely unstable ABL structure were observed on 22 July (the Haze I period) and 24 July, respectively, and the ambient particles sharply decreased before the outbreak stage. This emphasized that the ABL structure extremely restrained the diffusion of substances and was a prerequisite for pollution outbreaks. With clean and strong winds passing through Beijing, the ABL became unstable with a negative $\partial\theta/\partial z$ value and an increased ABLH. The high turbulence activity promoted pollution diffusion. Regardless of the wet deposition process or the high turbulence activity, air pollution would eventually dissipate once the atmosphere was in a specific state. The self-cleaning capacity of the atmosphere was responsible for air pollution diffusion. When the atmosphere is in a specific state, its self-cleaning capacity becomes dominant, which is worthy of further study.

**Data availability**

The surface $PM_{2.5}$ and $PM_{10}$ data and observation data of the other trace gases in this study can be accessed at http://106.37.208.233:20035/. Atmospheric reanalysis data were obtained from the National Centers for Environmental Prediction (NCEP) (https://www.esrl.noaa.gov/psd/data/). The other datasets can be obtained upon request from the corresponding author.

**Author contribution**

ZD and LG performed the research and wrote the paper, contributing equally to this study. XJ, QJ, WY and WX provided writing guidance, revised and polished the paper. LZ, TG, HB and WL designed the experiments and DL, MY, WX and WF carried them out. GC contributed to discussions of results. All the authors have made substantial contributions to the work reported in the manuscript.

**Competing interests.**

The authors declare that they have no conflict of interest.

**Acknowledgments**

This study was supported by the Ministry of Science and Technology of China (grant number 2016YFC0202001), the CAS Strategic Priority Research Program (XDA23020301) and the National Natural Science Foundation of China (grant number 41375036). The authors are grateful for services rendered by the National Oceanic and Atmospheric Administration (NOAA) and National Centers for Environmental Prediction (NCEP). The authors are thankful for the data support from the National Earth System Science Data Sharing Infrastructure, National Science & Technology Infrastructure of China (available at http://www.geodata.cn).